# MAX-SLICED BURES DISTANCE FOR INTERPRETING DISCREPANCIES

## ABSTRACT

We propose the max-sliced Bures distance, a lower bound on the max-sliced Wasserstein-2 distance, to identify the instances associated with the maximum discrepancy between two samples. The max-slicing can be decomposed into two asymmetric divergences each expressed in terms of an optimal slice or equivalently a 'witness' function that has large magnitude evaluations on a localized subset of instances in one distribution versus the other. We show how witness functions can be used to detect and correct for covariate shift through reweighting and to evaluate generative adversarial networks. Unlike heuristic algorithms for the max-sliced Wasserstein-2 distance that may fail to find the optimal slice, we detail a tractable algorithm that finds the global optimal slice and scales to large sample sizes. As the Bures distance quantifies differences in covariance, we generalize the max-sliced Bures distance by using non-linear mappings, enabling it to capture changes in higher-order statistics. We explore two types of non-linear mappings: positive semidefinite kernels where the witness functions belong to a reproducing kernel Hilbert space, and task-relevant mappings corresponding to a neural network. In the context of samples of natural images, our approach provides an interpretation of the Fréchet Inception distance by identifying the synthetic and natural instances that are either over-represented or under-represented with respect to the other sample. We apply the proposed measure to detect imbalances in class distributions in various data sets and to critique generative models.

## 1 INTRODUCTION

Divergence measures quantify the dissimilarity between probability distributions. They are fundamental to hypothesis testing and the estimation and criticism of statistical models, and serve as cost functions for optimizing generative adversarial neural networks (GANs). Although a multitude of divergences exists, not all of them are interpretable. A divergence is interpretable if can be expressed in terms of a real-valued *witness function* $\omega(\cdot)$ whose level-sets identify the specific subsets that are not well matched between the distributions, specifically, subsets which have much higher or much lower probability under one distribution versus the other. Localizing these discrepancies is useful for understanding and compensating for differences between two samples or distributions, to detect covariate shift (Shimodaira, 2000; Quionero-Candela et al., 2009; Lipton et al., 2018) or to evaluate generative models (Heusel et al., 2017).

While many divergences can be posed in terms of witness functions, not all witness functions are readily obtained or interpreted. From an information-theoretic perspective, the most natural witness function is the logarithm of the ratio of the densities (Kullback & Leibler, 1951) as in the Kullback-Leibler divergence. Applying other convex functions to the density ratio constitutes the family of $f$-divergences (Ali & Silvey, 1966; Rényi, 1961), which include the Hellinger, Jensen-Shannon, and others. However, without a parametric model estimating the densities from samples is challenging (Vapnik, 2013). Following Vapnik's advice to "try to avoid solving a more general problem as an intermediate step," previous work has sought to directly model the density ratio via kernel learning (Nguyen et al., 2008; Kanamori et al., 2009; Yamada et al., 2011; 2013; Saito et al., 2018; Lee et al., 2019) or to estimate an $f$-divergence by optimizing a function from a suitable family (Nguyen et al., 2010) such as a neural network Nowozin et al. (2016).

Witness functions need not rely on the density ratio. A wide class of divergences called integral probability metrics (IPMs) (Müller, 1997), which include total variation, the Wasserstein-1 distance, maximum mean discrepancy (MMD) (Gretton et al., 2007), and others (Mroueh et al., 2017), seek a witness function that maximizes the distance between the first moments of the witness function evaluations. In these cases the optimal witness function $\omega_\star(\cdot)$ has a greater expectation in one distribution compared to the other distribution. An IPM between two measures $\mu$ and $\nu$ is expressed as $\sup_{\omega \in \mathcal{F}} |\mathbb{E}_{X \sim \mu}[\omega(X)] - \mathbb{E}_{Y \sim \nu}[\omega(Y)]|$ for a given family of functions $\mathcal{F}$.

A class of related divergences are the max-sliced Wasserstein-$p$ distances, which seek a linear (Deshpande et al., 2019) or non-linear slicing function (Kolouri et al., 2019) that maximizes the Wasserstein-$p$ distance between the witness function evaluations for the two distributions. However, there are two difficulties with computing the max-sliced Wasserstein distance for two samples. The first is that it is a saddlepoint optimization problem, whose objective evaluation requires sorting the samples. Previous work has sought to approximate it using a first moment approximation (Deshpande et al., 2019) or to use a finite number of steps of a local optimizer (Kolouri et al., 2019), without any guarantee of obtaining an optimal witness function. Another difficulty is in the interpretation of the obtained witness function. Unlike the density ratio, there is no notion of whether the witness function will take higher values for points associated to one distribution versus the other. To address both of these issues we propose a max-sliced distance that replaces the Wasserstein-2 distance with a second-moment approximation based on the Bures distance (Dowson & Landau, 1982; Gelbrich, 1990). The Bures distance (Bures, 1969; Uhlmann, 1976) is a distance metric between positive semidefinite operators. It is well-known in quantum information theory (Nielsen & Chuang, 2000; Koltchinskii & Xia, 2015) and machine learning (Brockmeier et al., 2017; Muzellec & Cuturi, 2018; Zhang et al., 2020; Oh et al., 2020; De Meulemeester et al., 2020).

## 1.1 CONTRIBUTION

We propose a novel *IPM-like* divergence measure, the "max-sliced Bures distance", to identify localized regions and instances associated with the maximum discrepancy between two samples. The distance is expressed as the maximal difference between the *root mean square* (RMS) of the witness function evaluations $\sup_{\omega \in \mathcal{S}} \left| \sqrt{\mathbb{E}_{X \sim \mu}[\omega^2(X)]} - \sqrt{\mathbb{E}_{Y \sim \nu}[\omega^2(Y)]} \right|$, where $\mathcal{S}$ is an appropriate family of functions. As $|\Delta| = \max\{\Delta, -\Delta\}$, the max-sliced Bures can be expressed as the maximum of *one-sided* max-sliced divergences with optimal witness functions, $\omega_{\mu > \nu} = \arg\max_{\omega \in \mathcal{S}} \sqrt{\mathbb{E}_\mu[\omega^2(X)]} - \sqrt{\mathbb{E}_\nu[\omega^2(Y)]}$, and $\omega_{\mu < \nu} = \arg\max_{\omega \in \mathcal{S}} \sqrt{\mathbb{E}_\nu[\omega^2(Y)]} - \sqrt{\mathbb{E}_\mu[\omega^2(X)]}$. If the distributions are not well-matched, then $\omega_{\mu > \nu}$ has large magnitude function evaluations under a 'localized' subset of $\mu$ and smaller magnitude values for $\nu$, and the opposite for $\omega_{\mu < \nu}$. The two samples $\{x_i\}_{i=1}^m, \{y_i\}_{i=1}^n$ can be sorted by the magnitude of the witness function evaluations.[1]

Crucially, we detail a tractable optimization procedure that is guaranteed to yield a global optimum witness function for the one-sided max-sliced Bures divergence. When $\mathcal{X} = \mathbb{R}^d$ and the first or second moments distinguish the distributions, linear witness functions can be used $\mathcal{S} = \{\omega(\cdot) = \langle \cdot, \mathbf{w} \rangle : \mathbf{w} \in \mathbb{S}^{d-1}\}$, where $\mathbb{S}^{d-1}$ denotes the unit sphere in $\mathbb{R}^d$. The optimal witness function for the one-sided max-sliced Bures divergence $\omega_{\mu > \nu}(\cdot) = \langle \cdot, \mathbf{w}_{\mu > \nu} \rangle$ coincides with the subspace with the greatest difference in RMS, $\mathbf{w}_{\mu > \nu} = \arg\max_{\mathbf{w} \in \mathbb{S}^{d-1}} \sqrt{\mathbf{w}^\top \mathbb{E}[X X^\top] \mathbf{w}} - \sqrt{\mathbf{w}^\top \mathbb{E}[Y Y^\top] \mathbf{w}}$. This optimization problem depends on the dimension $d$; after computation of the covariance matrices, it is independent of the sample sizes $m \geq n$. In comparison, the optimal slice for the max-sliced Wasserstein may not be obtained, and even gradient ascent to a local optimum requires $\mathcal{O}(m \log m)$ at each function/gradient evaluation. Furthermore, the slice that maximizes the max-sliced Wasser-

---

[1]Four groups of 'witness points' (top-$K$ instances) can be inspected to identify any discrepancies:

$$\underbrace{\omega_{\mu > \nu}^2(x_{\mathring{\pi}(1)}) \geq \cdots \geq \omega_{\mu > \nu}^2(x_{\mathring{\pi}(K)})}_{\mathring{\pi} \text{ sorts } \{x_i\}_{i=1}^m \text{ to reveal examples from } \hat{\mu} \text{ with large } \omega_{\mu > \nu}^2} \quad \widetilde{\gg} \quad \underbrace{\omega_{\mu > \nu}^2(y_{\mathring{\sigma}(1)}) \geq \cdots \geq \omega_{\mu > \nu}^2(y_{\mathring{\sigma}(K)})}_{\mathring{\sigma} \text{ sorts } \{y_i\}_{i=1}^n \text{ to find the examples from } \hat{\nu} \text{ with large } \omega_{\mu > \nu}^2} \quad , \qquad (1)$$

$$\underbrace{\omega_{\mu < \nu}^2(x_{\acute{\pi}(1)}) \geq \cdots \geq \omega_{\mu < \nu}^2(x_{\acute{\pi}(K)})}_{\acute{\pi} \text{ sorts } \{x_i\}_{i=1}^m \text{ to find examples from } \hat{\mu} \text{ with large } \omega_{\mu < \nu}^2} \quad \widetilde{\ll} \quad \underbrace{\omega_{\mu < \nu}^2(y_{\acute{\sigma}(1)}) \geq \cdots \geq \omega_{\mu < \nu}^2(y_{\acute{\sigma}(K)})}_{\acute{\sigma} \text{ sorts } \{y_i\}_{i=1}^n \text{ to find examples from } \hat{\nu} \text{ with large } \omega_{\mu < \nu}^2} \quad , \qquad (2)$$

where $\mathring{\pi}, \acute{\pi}, \mathring{\sigma}, \acute{\sigma}$ denote permutations and $\widetilde{\gg}$ and $\widetilde{\ll}$ denote expected inequalities with a large difference.

Figure 1: The magnitude of the witness functions obtained from max-sliced Bures indicate discrepancies. (Left) A Gaussian kernel is used to construct non-linear witness functions and identify witness points. (Right) Witness points for real and fake images from stacked MNIST and CIFAR10. In each of the 6 frames, instances corresponding to the left hand sides of equation 1 and equation 2 are on the top; the bottom instances correspond to the right hand sides of the expected inequalities.

stein lacks an intrinsic ordering, and it is left to the user to determine whether instances from $\mu$ or $\nu$ have high or low values or magnitudes.

As second-order moments may be insufficient for distinguishing the distributions, we explore non-linear mappings of the random variables. Firstly, we consider a reproducing kernel Hilbert space (RKHS) $\mathcal{H}$ with the family of witness functions $\mathcal{S} = \{\omega(\cdot) = \langle \phi(\cdot), \omega \rangle_{\mathcal{H}} : \omega, \phi(\cdot) \in \mathcal{H}, \quad \langle \omega, \omega \rangle_{\mathcal{H}} = 1\}$. An example with Gaussian kernels is shown in Figure 1. Secondly, we use a pre-trained neural network to create a task-relevant mapping, computing the second-order statistics of the hidden-layer activations, and apply this in the context of samples of natural images. This enables interpretation of the Fréchet Inception distance (FID) (Heusel et al., 2017) by identifying the subspace and images associated with discrepancies between synthetic and natural images. We prove that the max-sliced Bures distance provides a lower bound on the max-sliced Fréchet distance.

Because of their similarity, we develop the max-sliced Bures distance in the context of max-sliced versions of the total variation and Wasserstein-2 distances. The kernel-based versions of these are novel contributions themselves. The max-sliced total variation distance is a special case of the covariance feature matching proposed by Mroueh et al. (2017).

In experimental results, we show applications of the linear and kernel-based versions to detect imbalances in class distributions of natural images and to critique GANs. We compare to other divergences expressed in terms of witness functions including MMD. Finally, we propose algorithms to reweight an empirical distribution in order to minimize max-sliced divergences (with applications to generating conditional distributions and covariate shift correction).

## 2 METHODOLOGY

Consider a topological space $\mathcal{X}$, a Borel $\sigma$-algebra $\mathcal{B}_{\mathcal{X}}$, and the set $\mathrm{Pr}(\mathcal{X})$ of Borel probability measures on $\mathcal{X}$. Let $\mu, \nu \in \mathrm{Pr}(\mathcal{X})$ denote two probability measures, and $X \sim \mu$ and $Y \sim \nu$ be two random variables $X, Y \in \mathcal{X}$. Let $\kappa$ denote a positive-definite, bounded kernel function $\kappa : \mathcal{X} \times \mathcal{X} \to B \subset \mathbb{R}$. For any $\kappa$, there is an implicit mapping $\phi : \mathcal{X} \to \mathcal{H}$ that maps any element $x \in \mathcal{X}$ to an element in the reproducing kernel Hilbert space (RKHS) $\phi(x) \in \mathcal{H}$ such that $\kappa(x, y) = \langle \phi(x), \phi(y) \rangle_{\mathcal{H}} = \langle \kappa(\cdot, x), \kappa(\cdot, y) \rangle_{\mathcal{H}}$ for $x, y \in \mathcal{X}$, and $\forall \omega \in \mathcal{H}, \omega(x) = \langle \omega, \kappa(\cdot, x) \rangle_{\mathcal{H}}$ (Aronszajn, 1950). $\forall \omega \in \mathcal{H}, \|\omega\|_2 = \sqrt{\langle \omega, \omega \rangle_{\mathcal{H}}}$. When clear, we drop the $\mathcal{H}$ subscript on the inner product. A rank-1 RKHS operator is denoted as $\omega \otimes \psi \in \mathcal{H} \times \mathcal{H}$ with $\langle (\omega \otimes \psi) \phi(x), \phi(y) \rangle = \langle \psi, \phi(x) \rangle \langle \omega, \phi(y) \rangle = \psi(x) \omega(y)$ for $x, y \in \mathcal{X}$. Denote by $m_X = \mathbb{E}_{X \sim \mu}[\phi(X)] \in \mathcal{H}$ and $m_Y = \mathbb{E}_{Y \sim \nu}[\phi(Y)] \in \mathcal{H}$ the first moments of the random variables in the RKHS. The uncentered second moments are $\rho_X = \mathbb{E}[\phi(X) \otimes \phi(X)] \in \mathcal{H} \times \mathcal{H}$ and $\rho_Y = \mathbb{E}[\phi(Y) \otimes \phi(Y)] \in \mathcal{H} \times \mathcal{H}$. The covariance operators are $\Sigma_X = \rho_X - m_X \otimes m_X$ and $\Sigma_Y = \rho_Y - m_Y \otimes m_Y$.

## 2.1 Divergences as Distance Metrics

Let $D(\mu,\nu)$ denote a divergence $D : \Pr(\mathcal{X}) \times \Pr(\mathcal{X}) \to [0,\infty)$. It is a distance metric between measures (a probability metric) if all of the following statements hold: (i) $\mu = \nu \implies D(\mu,\nu) = 0$, (ii) $D(\mu,\nu) = 0 \implies \mu = \nu$, (iii) $D(\mu,\nu) = D(\nu,\mu)$, (iv) $D(\mu,\nu) \leq D(\mu,\xi) + D(\nu,\xi)$. It is a semi-metric if all properties aside from (ii) hold. Müller (1997) defines the class of integral probability metrics as the supremum of the absolute difference between expectations

$$D_{\mathcal{F}}(\mu,\nu) = \sup_{\omega \in \mathcal{F}} \left| \int_{\mathcal{X}} \omega(x)d\mu(x) - \int_{\mathcal{X}} \omega(x)d\nu(x) \right| = \sup_{\omega \in \mathcal{F}} \left| \mathbb{E}[\omega(X)] - \mathbb{E}[\omega(Y)] \right|.$$

With appropriate choice of the family of functions $\mathcal{F}$, this form yields well-known divergences (Sriperumbudur et al., 2010), e.g., when $\mathcal{F}$ is the set of functions with Lipschitz constant less than 1, the resulting divergence is the Wasserstein-1 distance metric. Another example of an IPM is when $\mathcal{F} = \{\omega \in \mathcal{H} : \|\omega\|_2 \leq 1\}$, which yields MMD (Gretton et al., 2007), defined as

$$D_{MMD}^{\mathcal{H}}(\mu,\nu) = \sup_{\omega \in \mathcal{H}: \|\omega\|_2 \leq 1} \{\mathbb{E}[\omega(X)] - \mathbb{E}[\omega(Y)] = \mathbb{E}[\langle \phi(X) - \phi(Y), \omega \rangle]\} = \|m_X - m_Y\|_2$$

$$= \sqrt{\mathbb{E}_{X\sim\mu,X'\sim\mu}[\kappa(X,X')] + \mathbb{E}_{Y\sim\nu,Y'\sim\nu}[\kappa(Y,Y')] - 2\mathbb{E}_{X\sim\mu,Y\sim\nu}[\kappa(X,Y)]}. \quad (3)$$

For characteristic kernels such as the Laplacian and Gaussian kernels, the mean embedding $\mathbb{E}_{X\sim\mu}[\phi(X)] : \Pr(\mathcal{X}) \to \mathcal{H}$ is an injective function (Sriperumbudur et al., 2008; Fukumizu et al., 2009; Sriperumbudur et al., 2010), capturing the full statistics of $\mu$. In these cases, MMD is a distance metric on $\Pr(\mathcal{X})$; likewise, distance metrics between the operators $\rho_X = \mathbb{E}[\phi(X) \otimes \phi(X)]$ and $\rho_Y = \mathbb{E}[\phi(Y) \otimes \phi(Y)]$ induce probability metrics for characteristic kernels (Zhang et al., 2020).

## 2.2 Operator Distances for Defining Divergences

Total variation (TV) is a well-known probability metric and an integral probability metric (Müller, 1997), taking the form $(1/2)\sum_i |p_i - q_i|$ for discrete measures, for which $p_i = \mu(x_i)$ and $q_i = \nu(x_i)$ where $\{x_i\}_i = \mathcal{X}$. The TV distance between operators in the RKHS is a divergence

$$D_{TV}^{\mathcal{H}}(\mu,\nu) \triangleq d_{TV}(\rho_X, \rho_Y) \triangleq \frac{1}{2}\|\rho_X - \rho_Y\|_1, \quad (4)$$

where $\|\cdot\|_1$ denotes the trace norm (Schatten 1-norm), which is the sum of the singular values.

The Bures distance generalizes the Hellinger distance $\sqrt{(1/2)\sum_i(\sqrt{p_i} - \sqrt{q_i})^2}$ to positive semidefinite operators (Fuchs & Van De Graaf, 1999; Bromley et al., 2014; Bhatia et al., 2019). The kernel Bures divergence $D_B^{\mathcal{H}}(\mu,\nu)$ and the Bures distance $d_B(\rho_X, \rho_Y)$ are defined as

$$D_B^{\mathcal{H}}(\mu,\nu) = d_B(\rho_X, \rho_Y) \triangleq \sqrt{\|\rho_X\|_1 + \|\rho_Y\|_1 - 2\|\sqrt{\rho_X}\sqrt{\rho_Y}\|_1}. \quad (5)$$

The Bures distance is used to define the Wasserstein-2 (W2) distance between Gaussian measures, i.e., the Fréchet distance (Fréchet, 1957; Dowson & Landau, 1982). The multivariate Fréchet distance provides a lower bound for the W2 distance (Gelbrich, 1990).[2] The kernel Gauss-Wasserstein distance (Zhang et al., 2020; Oh et al., 2020) is defined as

$$D_{GW}^{\mathcal{H}}(\mu,\nu) \triangleq \sqrt{\|m_X - m_Y\|_2^2 + d_B^2(\Sigma_X, \Sigma_Y)} = \sqrt{[D_{MMD}^{\mathcal{H}}(\mu,\nu)]^2 + d_B^2(\Sigma_X, \Sigma_Y)}. \quad (6)$$

Zhang et al. (2020) also proposed the kernel Wasserstein-$p$ distance between $\mu$ and $\nu$,

$$W_p^{\mathcal{H}}(\mu,\nu) \triangleq \left( \inf_{\gamma \in \Gamma(\mu,\nu)} \mathbb{E}_{(X,Y)\sim\gamma}[d_\kappa^p(X,Y)] \right)^{\frac{1}{p}}, \quad p \geq 1, \quad (7)$$

where $\Gamma(\mu,\nu)$ defines the set of all joint distributions coupling $\mu$ and $\nu$, and $d_\kappa^p(X,Y) = \|\phi(X) - \phi(Y)\|_2^p$. For $p = 2$, $d_\kappa^2(X,Y) = \kappa(X,X) + \kappa(Y,Y) - 2\kappa(X,Y)$. When $p = 2$ and $\phi(x) \mapsto x \in \mathbb{R}^d$ such that $\mathcal{H} = \mathbb{R}^d$, the standard W2 distance $W_2^{\mathbb{R}^d}(\mu,\nu)$ is obtained.

---

[2]In the finite-dimensional case, the multivariate Fréchet distance (squared) is often expressed as $\|\mathbf{m}_X - \mathbf{m}_Y\|_2^2 + \mathrm{tr}(\Sigma_X + \Sigma_Y - 2\sqrt{\Sigma_X \Sigma_Y})$; the trace term is the squared Bures distance $d_B^2(\Sigma_X, \Sigma_Y) = \mathrm{tr}(\Sigma_X + \Sigma_Y) - 2\|\sqrt{\Sigma_X}\sqrt{\Sigma_Y}\|_1$, where $\|\sqrt{\Sigma_X}\sqrt{\Sigma_Y}\|_1 = \mathrm{tr}(\sqrt{\Sigma_X \Sigma_Y})$ (Dowson & Landau, 1982).

### 2.3 DIVERGENCES BASED ON SLICING HILBERT SPACES

The sliced Wasserstein distance (Wu et al., 2019; Deshpande et al., 2018; Kolouri et al., 2018), and max-sliced Wasserstein distance (Deshpande et al., 2019; Kolouri et al., 2019) evaluate discrepancies in linear or non-linear one-dimensional subspaces. A motivation for this is the analytic solution of the Wasserstein-$p$ distance in one dimension. The max-sliced Wasserstein-$p$ distance takes the form $max\text{-}W_p^{\mathbb{R}^d}(\mu, \nu) \propto \sup_{\mathbf{w} \in \mathbb{S}^{d-1}} \inf_{\gamma \in \Gamma(\mu,\nu)} \mathbb{E}_{(X,Y) \sim \gamma}[\langle X - Y, \mathbf{w} \rangle^p], \quad p \geq 1$. Similarly, we propose the max-sliced Bures, the kernel TV, and the kernel Wasserstein-$p$ distances using the rank-1 operator $\Omega = \omega \otimes \omega \in \mathcal{H} \times \mathcal{H}$, which projects (slices) the RKHS along a one-dimensional subspace defined by the ray $\omega \in \mathcal{H}$, with $\langle \omega, \phi(X) \rangle = \omega(X)$, due to the reproducing property. In this formulation, $\omega : \mathcal{X} \to \mathbb{R}$ is the witness function from the set $\mathcal{S} = \{\omega \in \mathcal{H} : \|\omega\|_2 = 1\}$. Notably, a linear slice in the RKHS is a possibly non-linear function in the input space.

For conciseness, we denote the mean square witness function evaluations $\mathbb{E}[\omega^2(X)] = \langle \omega, \rho_X \omega \rangle = \|\sqrt{\rho_X}\omega\|_2^2$ as $\|\omega\|_\mu^2$, and $\mathbb{E}[\omega^2(Y)] = \langle \omega, \rho_Y \omega \rangle = \|\sqrt{\rho_Y}\omega\|_2^2$ as $\|\omega\|_\nu^2$. The RMS $\|\omega\|_\mu$ is an $L_2$ semi-norm on $\omega$ induced by the positive semidefinite operator $\sqrt{\rho_X}$. The max-sliced kernel TV, Bures, and W2 distances, derived in appendix A.1, are expressed as

$$max\text{-}D_{TV}^{\mathcal{H}}(\mu, \nu) \triangleq \frac{1}{2} \max \left\{ \sup_{\omega \in \mathcal{S}} \|\omega\|_\mu^2 - \|\omega\|_\nu^2, \quad \sup_{\omega \in \mathcal{S}} \|\omega\|_\nu^2 - \|\omega\|_\mu^2 \right\}, \tag{8}$$

$$max\text{-}D_{B}^{\mathcal{H}}(\mu, \nu) \triangleq \max \left\{ \sup_{\omega \in \mathcal{S}} \|\omega\|_\mu - \|\omega\|_\nu, \quad \sup_{\omega \in \mathcal{S}} \|\omega\|_\nu - \|\omega\|_\mu \right\}, \text{ and} \tag{9}$$

$$max\text{-}W_2^{\mathcal{H}}(\mu, \nu) \triangleq \sup_{\omega \in \mathcal{S}} \sqrt{\|\omega\|_\mu^2 + \|\omega\|_\nu^2 - \sup_{\gamma \in \Gamma(\mu,\nu)} \mathbb{E}_{(X,Y) \sim \gamma} [2\omega(X)\omega(Y)]}, \tag{10}$$

respectively. The inner supremums in equation 8 and equation 9 are the one-sided divergences.

The max-sliced TV distance is an IPM with $\mathcal{F} = \{\omega^2(x) = \langle \phi(x), \omega \rangle^2 : \forall \omega \in \mathcal{S}\}$ and is a special case of the IPM$_\Sigma$ divergence proposed by Mroueh et al. (2017). While *not* an IPM, the max-sliced kernel Bures distance can be directly related to max-sliced versions of the TV, Fréchet, and W2 distance, as detailed by the following results.

**Theorem 1.** *The square of the max-sliced Bures distance in the RKHS $\mathcal{H}$ is less than or equal to twice the max-sliced TV distance, $max\text{-}D_B^{\mathcal{H}}(\mu, \nu) \leq \sqrt{2\left(max\text{-}D_{TV}^{\mathcal{H}}(\mu, \nu)\right)}$.*

**Theorem 2.** *The max-sliced Bures distance in the RKHS $\mathcal{H}$ is a lower bound on the kernel max-sliced Gauss-Wasserstein distance, $max\text{-}D_B^{\mathcal{H}}(\mu, \nu) \leq max_L\text{-}D_{GW}^{\mathcal{H}}(\mu, \nu) \leq max_U\text{-}D_{GW}^{\mathcal{H}}(\mu, \nu)$.*

**Theorem 3.** *The max-sliced Bures distance in the RKHS is a lower bound on the kernel max-sliced W2 distance, $max\text{-}D_B^{\mathcal{H}}(\mu, \nu) \leq max\text{-}W_2^{\mathcal{H}}(\mu, \nu)$.*

These results trivially translate to the linear kernel case $\mathcal{H} = \mathbb{R}^d$, $\mathcal{S} = \mathbb{S}^{d-1}$, $\omega(\mathbf{x}) = \langle \mathbf{w}, \mathbf{x} \rangle$, for $\mathbf{w}, \mathbf{x} \in \mathbb{R}^d$ and $\mathcal{X} \subseteq \mathbb{R}^d$. The latter two show that the max-sliced Bures distance is a lower-bound on the max-sliced Fréchet distance, which is a lower-bound on the max-sliced W2 distance. The proofs and other relationships among the divergences are in Appendix A.2.

### 2.4 COMPUTING THE MAX-SLICED DIVERGENCES

The max-sliced kernel Bures, TV, and W2 distances require solving optimization problems to find the optimal witness function. As noted by others investigating IPMs (Mroueh et al., 2017; Li et al., 2017; Kolouri et al., 2019), the witness function can be defined using a family of functions implemented as a neural network. In this context, Goodfellow et al. (2014) use the divergence $D_A(\mu, \nu) = \max_\omega \mathbb{E}[\log \omega(X)] + \mathbb{E}[\log(1 - \omega(Y))]$, where $\omega : \mathcal{X} \to (0, 1)$. In comparison, the Wasserstein-1 distance $D_{W^1}(\mu, \nu) = \sup_{\omega \in \text{Lip}^1} \mathbb{E}[\omega(X)] - \mathbb{E}[\omega(Y)]$ requires a Lipschitz con-

straint (Arjovsky et al., 2017; Gulrajani et al., 2017).[3] Table 1 compares the form and constraints.

Table 1: Divergences written in terms of witness functions. Closed-form solutions denoted $\omega^*$.

| | |
|---|---|
| $D_A(\mu, \nu)$ | $= \max_{\omega:\omega(\cdot)\in(0,1)} \mathbb{E}[\log \omega(X)] + \mathbb{E}[\log(1-\omega(Y))], \quad \omega^*(\cdot) = \frac{d\mu(\cdot)}{d\mu(\cdot)+d\nu(\cdot)}.$ |
| $D_{W^1}(\mu, \nu)$ | $= \sup_{\omega\in\text{Lip}^1} \mathbb{E}[\omega(X)] - \mathbb{E}[\omega(Y)].$ |
| $D_{MMD}^{\mathcal{H}}(\mu, \nu)$ | $= \sup_{\omega\in\mathcal{H}:\|\omega\|_2\leq 1} \mathbb{E}[\omega(X)] - \mathbb{E}[\omega(Y)], \quad \omega^* = \frac{m_X - m_Y}{\|m_X - m_Y\|_2}.$ |
| $max\text{-}D_{TV}^{\mathcal{H}}(\mu, \nu)$ | $= \sup_{\omega\in\mathcal{H}:\|\omega\|_2\leq 1} \frac{1}{2}\left|\mathbb{E}[\omega^2(X)] - \mathbb{E}[\omega^2(Y)]\right|.$ |
| $max\text{-}D_B^{\mathcal{H}}(\mu, \nu)$ | $= \sup_{\omega\in\mathcal{H}:\|\omega\|_2\leq 1} \left|\sqrt{\mathbb{E}[\omega^2(X)]} - \sqrt{\mathbb{E}[\omega^2(Y)]}\right|.$ |

## 2.5 SAMPLE-BASED ESTIMATORS FOR MAX-SLICED DIVERGENCES

We consider the case of finite samples expressed as empirical measures $\hat{\mu} = \sum_{i=1}^m \mu_i \delta_{x_i}$ and $\hat{\nu} = \sum_{i=1}^n \nu_i \delta_{y_i}$ for the samples $\{x_i\}_{i=1}^m$ and $\{y_i\}_{i=1}^n$ with discrete probability masses denoted as column vectors $[\mu_1, \ldots, \mu_m]^\top = \boldsymbol{\mu} \in [0,1]^m$, $\langle \boldsymbol{\mu}, \mathbf{1} \rangle = 1$, and $[\nu_1, \ldots, \nu_n]^\top = \boldsymbol{\nu} \in [0,1]^n$, $\langle \boldsymbol{\nu}, \mathbf{1} \rangle = 1$. The kernel-based max-sliced divergences optimize the witness function $\omega(\cdot) = \sum_{i=1}^l \alpha_i \kappa(\cdot, z_i)$ in terms of the dual variables $\boldsymbol{\alpha} \in \mathbb{R}^l$ corresponding to a subset of the pooled sample $\{x_i\}_{i=1}^m \cup \{y_i\}_{i=1}^n$. The optimization problems and algorithms are detailed in appendix A.4.

For clarity, we proceed to the linear kernel case for a finite-dimensional embedding $\phi(x) = \mathbf{x} \in \mathbb{R}^d$. After embedding, kernel evaluations correspond to vector inner-products $\kappa(x, y) = \langle \phi(x), \phi(y) \rangle = \mathbf{x}^\top \mathbf{y}$. Let $\mathbf{X} = [\mathbf{x}_1, \ldots, \mathbf{x}_m] \in \mathbb{R}^{d\times m}$ and $\mathbf{Y} = [\mathbf{y}_1, \ldots, \mathbf{y}_n] \in \mathbb{R}^{d\times n}$ denote the sample points with corresponding masses $\boldsymbol{\mu}$ and $\boldsymbol{\nu}$, respectively. The witness function is the inner product $\omega(x) = \mathbf{w}^\top \mathbf{x}$, where the variable $\mathbf{w}$ defines the slice with $\|\omega\|_{\hat{\mu}}^2 = \mathbf{w}^\top \boldsymbol{\rho}_X \mathbf{w}$ with $\boldsymbol{\rho}_X = \mathbf{X}\mathbf{D}_{\boldsymbol{\mu}}\mathbf{X}^\top$ and $\|\omega\|_{\hat{\nu}}^2 = \mathbf{w}^\top \boldsymbol{\rho}_Y \mathbf{w}$ with $\boldsymbol{\rho}_Y = \mathbf{Y}\mathbf{D}_{\boldsymbol{\nu}}\mathbf{Y}^\top$, where $\mathbf{D}_{\mathbf{v}}$ is diagonal with entries $\mathbf{v}$. For i.i.d. samples, $\mathbf{D}_{\boldsymbol{\mu}} = \frac{1}{m}\mathbf{I}$ and $\mathbf{D}_{\boldsymbol{\nu}} = \frac{1}{n}\mathbf{I}$. The max-sliced TV, Bures, and W2 divergences are

$$max\text{-}D_{TV}^{\mathbb{R}^d}(\hat{\mu}, \hat{\nu}) = \max_{\mathbf{w}:\|\mathbf{w}\|_2\leq 1} \left|\mathbf{w}^\top(\boldsymbol{\rho}_X - \boldsymbol{\rho}_Y)\mathbf{w}\right| = \lambda_1(\boldsymbol{\rho}_X - \boldsymbol{\rho}_Y), \tag{11}$$

$$max\text{-}D_B^{\mathbb{R}^d}(\hat{\mu}, \hat{\nu}) = \max_{\mathbf{w}:\|\mathbf{w}\|_2\leq 1} \left|\sqrt{\mathbf{w}^\top \boldsymbol{\rho}_X \mathbf{w}} - \sqrt{\mathbf{w}^\top \boldsymbol{\rho}_Y \mathbf{w}}\right|, \text{ and} \tag{12}$$

$$max\text{-}W_2^{\mathbb{R}^d}(\hat{\mu}, \hat{\nu}) = \max_{\mathbf{w}:\|\mathbf{w}\|_2\leq 1} \sqrt{\mathbf{w}^\top(\boldsymbol{\rho}_X + \boldsymbol{\rho}_Y)\mathbf{w} - 2\max_{\mathbf{P}\in\mathcal{P}_{\hat{\mu},\hat{\nu}}} \mathbf{w}^\top \mathbf{X}\mathbf{P}^\top \mathbf{Y}^\top \mathbf{w}}, \tag{13}$$

where $\lambda_1(\cdot)$ denotes the largest magnitude eigenvalue of the argument and $\mathcal{P}_{\hat{\mu},\hat{\nu}} = \{\mathbf{P} \in [0,1]^{m\times n} | \mathbf{P}\mathbf{1}_n = \boldsymbol{\mu}, \mathbf{P}^\top \mathbf{1}_m = \boldsymbol{\nu}\}$ is a transportation polytope.

These three optimizations differ in difficulty. The first two require only the sample means $\mathbf{m}_X, \mathbf{m}_Y$ and covariance matrices $\boldsymbol{\Sigma}_X, \boldsymbol{\Sigma}_Y$, since $\boldsymbol{\rho}_X = \mathbf{m}_X\mathbf{m}_X^\top + \frac{m-1}{m}\boldsymbol{\Sigma}_X$ and $\boldsymbol{\rho}_Y = \mathbf{m}_Y\mathbf{m}_Y^\top + \frac{n-1}{n}\boldsymbol{\Sigma}_Y$ (assuming unbiased covariance estimates). The one-sided max-sliced TV divergences can be solved by finding the eigenvectors associated to the largest eigenvalues of $\boldsymbol{\rho}_X - \boldsymbol{\rho}_Y$ and $\boldsymbol{\rho}_Y - \boldsymbol{\rho}_X$. Likewise the optimal slice for each one-sided max-sliced Bures divergence requires solving a series of eigenvector problems. Specifically, if $\boldsymbol{\rho}_X$ and $\boldsymbol{\rho}_Y$ are strictly positive definite, then the optimal witness function is $\omega_{\mu>\nu}(\cdot) = \langle \mathbf{w}_{\gamma^\star}, \cdot \rangle$, where $\gamma^\star \in (0,1]$ solves the optimization problem

$$\gamma^\star = \arg\max_{0<\gamma\leq 1} \sqrt{\mathbf{w}_\gamma^\top \boldsymbol{\rho}_X \mathbf{w}_\gamma} - \sqrt{\mathbf{w}_\gamma^\top \boldsymbol{\rho}_Y \mathbf{w}_\gamma}, \quad \mathbf{w}_\gamma = \arg\max_{\mathbf{w}:\|\mathbf{w}\|_2\leq 1} \mathbf{w}^\top(\gamma\boldsymbol{\rho}_X - \boldsymbol{\rho}_Y)\mathbf{w}. \tag{14}$$

The general case involves checking the nullspace of $\boldsymbol{\rho}_Y$ and is given in the Appendix A.5. In comparison, the max-sliced W2 distance is a saddlepoint optimization problem (Deshpande et al., 2019).

---

[3]In the context of GANs, the sum of the one-sided max-sliced Bures divergences may prove more appropriate for training, since it allows for separate witness functions for over- and under-representation. However, as its witness functions tend to localize discrepancies, even two witness functions may not make efficient use of the generator's samples. Instead, a distributional-version of the sliced Bures distance akin to the recent distributional-sliced Wasserstein proposal (Anonymous, 2021a) or the Bures distance itself (Anonymous, 2021b) could be used. Nonetheless, this localization property is what makes the max-sliced Bures interpretable.

Following Kolouri et al. (2019), gradient ascent on $\mathbf{w}$ can be performed with first-order solves. Each gradient evaluation requires solving the transport map by sorting $\mathbf{X}^\top \mathbf{w}$ and $\mathbf{Y}^\top \mathbf{w}$. For this, we use ADAM (Kingma & Ba, 2015) and quasi-Newton approaches, such as MINFUNC (Schmidt, 2012). The same approaches can be used to approximate $max\text{-}D_B$ after smoothing $\sqrt{\cdot}$ as $\sqrt{\cdot + 0.01}$.

## 3 EXPERIMENTS

We present various examples of using the proposed max-sliced divergences to identify the discrepancies between two samples. We apply the proposed approach to detect mismatched distributions of natural and fake images using the internal representation of the Inception Network (Szegedy et al., 2016) as in the Fréchet Inception distance (Heusel et al., 2017) and the Inception score (Salimans et al., 2016). We investigate whether the witness functions detect covariate shift caused by class imbalances. Then, we propose optimizing the weights $\boldsymbol{\nu}$ to compensate for covariate shift. Finally, we use the one-sided max-sliced Bures divergence to monitor mode dropping during GAN training.

### 3.1 INTERPRETING THE FRÉCHET INCEPTION DISTANCE

We use a linear witness function to identify instances that are not well matched between two samples of real or fake images represented by internal activations of the Inception object classifying network (Szegedy et al., 2016). Specifically, we search for witness functions with the form $\omega(x) = \langle \mathbf{w}, \phi(x) \rangle$, where the vector $\phi(x) \in \mathbb{R}^{2048}$ is an Inception code—the internal activations of penultimate layer of the network after pooling (Heusel et al., 2017).

Figure 2 shows the performance of the proposed measure to identify instances associated with imbalanced representation of particular classes. In particular, $\hat{\mu}$ is a uniform sample from the training set and $\hat{\nu}$ is a sample from the test set with less instances from one class. Using the one-sided max-sliced Bures divergence we obtain the optimal slice $\omega_{\mu>\nu}$ and apply it to the imbalanced sample $\hat{\nu}$, identifying the top-10 witness points with the largest magnitude witness function evaluations $\omega_{\mu>\nu}^2(y_{\dot{\sigma}(1)}) \geq \ldots \geq \omega_{\mu>\nu}^2(y_{\dot{\sigma}(10)})$, where $\dot{\sigma}$ is a permutation corresponding to sorting $\{y_i\}_{i=1}^n$ by descending magnitude. (While this may seem counterintuitive as it is expected that $\max_{1 \leq i \leq m} \omega_{\mu>\nu}^2(x_i) \gg \max_{1 \leq i \leq n} \omega_{\mu>\nu}^2(y_i)$, since $\omega_{\mu>\nu}$ corresponds to a one-dimensional subspace, $\{y_{\dot{\sigma}(i)}\}_{i=1}^K$ are the $K$ instances from $\hat{\nu}$ with the largest norm after projection to this subspace.) The performance is quantified by the precision of the labels of these instances (ideally, these witness points should be from the underrepresented class). Notably, the mean precision of the top-10 instances is 0.79 or better across the classes with a mean average precision (MAP) of 0.94 when the class probabilities differ by 2% (10.2% for majority and 8.2% for minority). This is compared to a MAP of 0.82 for the first-moment based surrogate of the max-sliced W2 distance (Deshpande et al., 2019). Computing the max-sliced W2 distance takes much longer to run on this sample size.

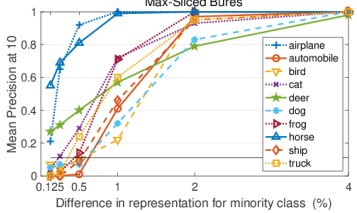 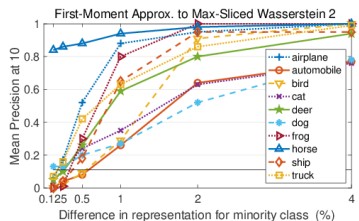

Figure 2: Max-sliced divergences using the Inception Network representation are applied to samples with mismatched class distributions in CIFAR10. The first sample consists of the training set (balanced classes with $m$=50,000), and the second sample is an imbalanced subset of the test set with $n$=10,000. Each curve is the mean precision@10 (averaged across 10 random draws) for test sets where the given class is subsampled at different levels of imbalance and other classes are balanced.

Next we generate a set of 50,000 synthetic images for an AutoGAN instance pre-trained on CIFAR10 (Gong et al., 2019), which has an Inception score of 8.525 and a FID score of 12.41. We applied both one-sided max-sliced Bures divergences to identify the two subspaces that maximize the difference in RMS between fake and real images. Figure 3 details the top-10 images in

each subspace and their realism scores $R$ (Kynkäänniemi et al., 2019).[4] Applying the max-sliced Wasserstein-2 distance yielded almost the same solution as $\mathbf{w}_{\text{Real}<\text{Fake}}$ (a linear correlation of 0.992).

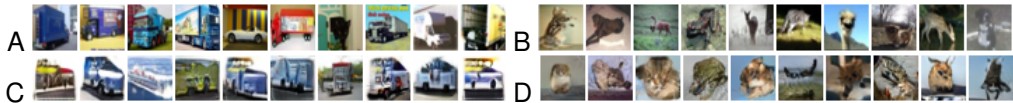

Figure 3: Max-slicing Inception codes to illustrate the AutoGAN discrepancies. One-sided max-sliced Bures is used to identify two witness function (as linear subspaces of the Inception codes) that differentiate the real from fake samples. (Left: A,C) Images in subspace under-represented by fake $\mathbf{w}_{\text{Real}>\text{Fake}}$. (Right: B,D) Images in subspace over-represented by fake $\mathbf{w}_{\text{Real}<\text{Fake}}$. (Top: A,B) Real CIFAR10 test images. (Bottom: C,D) Fake images. (C) Realism scores (median and range): 0.92 (0.84–1.03). (D) Realism scores (median and range): 0.68 (0.62–0.73)

## 3.2 BASELINE COMPARISON ON COVARIATE SHIFT DETECTION

We compare the proposed max-sliced Bures distance and the resulting max-sliced Fréchet distance to the max-sliced W2 distance for linear witness functions. Figure 4 compares the divergence estimates instances associated with a simple case of covariate shift with the MNIST data set. Notably, the precision of detecting class imbalances is higher for smaller samples using a kernel (Appendix A.8).

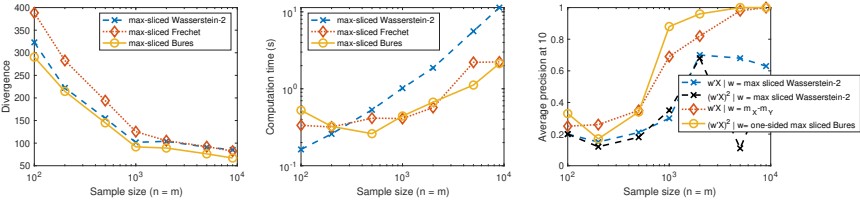

Figure 4: Max-sliced distances applied to two samples from MNIST. The first sample $\hat{\mu}$ is $m$ images drawn uniformly from the training set, and the second sample $\hat{\nu}$ is $n$ images from the test set where one digit is a minority class $l \in \{0, \ldots, 9\}$ with prevalence of 5%. (Left) Divergence estimates across sample size with $l = 7$. For $m < 2000$, gradient-based approaches for the max-sliced W2 distance fail to obtain the optimal slice as it should upper bound the max-sliced Fréchet distance. (Center) Computation time. (Right) Each curve is the average precision@10 (averaged across the 10 classes). The one-sided max-sliced Bures yields the witness function $\omega_{\mu>\nu}(\cdot) = \langle \mathbf{w}, \cdot \rangle$, which is applied to reliably identify the instances from $\hat{\mu}$ that are from the minority class for $m \geq 1000$.

## 3.3 COVARIATE SHIFT CORRECTION BY REWEIGHTING

We consider the task of reweighting the instances in one sample to minimize the max-sliced Bures distance. This optimization problem can be expressed as $\min_{\boldsymbol{\nu} \in \mathbb{R}^n_{\geq 0}: \sum_i \nu_i = 1} J(\boldsymbol{\nu})$, where $J(\boldsymbol{\nu}) \propto$ $max\text{-}D_B^{\mathbb{R}^d}(\hat{\mu}, \hat{\nu})$. As shown in appendix A.7, this is a convex minimization problem with a simplex constraint on $\boldsymbol{\nu}$. We apply the Frank-Wolfe algorithm (Jaggi, 2013) to iteratively adjust the weight of one instance at each iteration. The performance is quantified in terms of the Fréchet Inception distance between the real-test images of the class and the reweighted sample of fake images. For comparison, we also optimize reweightings that minimize the W2 distance and the max-sliced W2 distance (using 10 mini-batches of size $n = m = 100$ at each iteration). The average FID distance across the classes is 49.15 for the max-sliced Bures reweighting compared to 68.1 and 72.5 for the mini-batch W2 and max-sliced W2. (See Table 4 in the Appendix for full results).

---

[4]We also compute the realism scores of the entire set of fake images using the set of 10,000 test images and 3-nearest neighbor distances. The Spearman rank correlation between the realism scores for the full set of fake images and the witness function evaluations is -0.70 for $\omega^2_{\text{Real}<\text{Fake}}$ and it is 0.17 for $\omega^2_{\text{Real}>\text{Fake}}$. This means the realism score has a strong inverse correlation with $\omega^2_{\text{Real}<\text{Fake}} \propto -R$, and a weak correlation for $R \propto \omega^2_{\text{Real}>\text{Fake}}$.

Figure 5 shows results of a reweighting uniform distributions to match target distributions using either linear slices or random Fourier bases (Rahimi & Recht, 2008) for approximating a Gaussian kernel.[5] For the latter, using the max-sliced Bures as a loss achieves the lowest reweighted W2 distance, which is computed by solving a discrete transportation problem (Flamary & Courty, 2017).

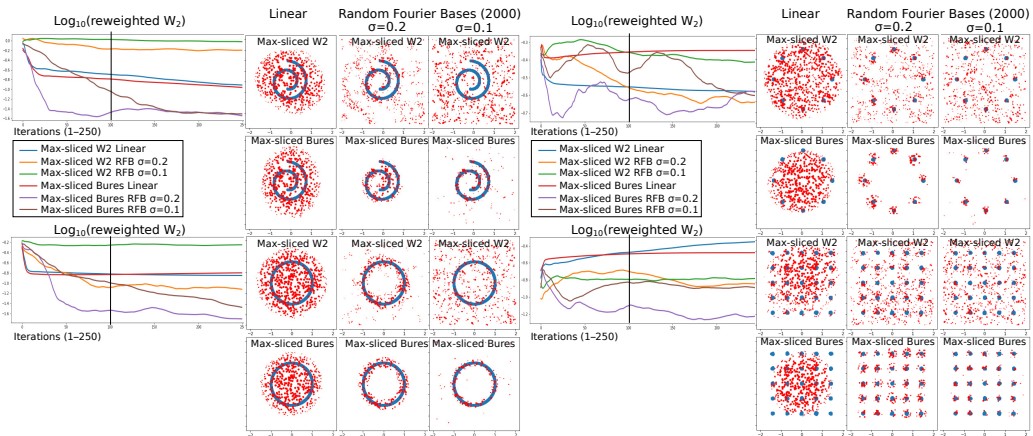

Figure 5: Reweighting a uniform distribution to match various target distributions by minimizing the max-sliced W2 distance or the max-sliced Bures distance with either a linear kernel or random Fourier bases ($d$=2,000, $\sigma \in \{0.1, 0.2\}$). Examples follow Kolouri et al. (2019) and uses ADAM (Kingma & Ba, 2015) defaults and a learning rate of $10^{-2}$. A point's size is proportional to their weights after 100 iterations. Learning curves are the weighted W2 distance (log-scale).

## 3.4 DETECTING MODE DROPPING

We create a 3-channel "Stacked MNIST" data set with 500,000 images from the MNIST training set to test mode dropping detection throughout GAN training (DCGAN architecture). The training set $\hat{\mu}$ has 1000 possible modes corresponding to all 3-digit combinations. At the end of each training epoch (1000 iterations), a fake sample $n = 10^4$ is generated. To verify mode coverage we use a 4-layer conv. net trained on single-channel MNIST; any missing combination of 3-digit labels is considered a dropped mode. Figure 6 details using the proposed approach to detect missing modes.

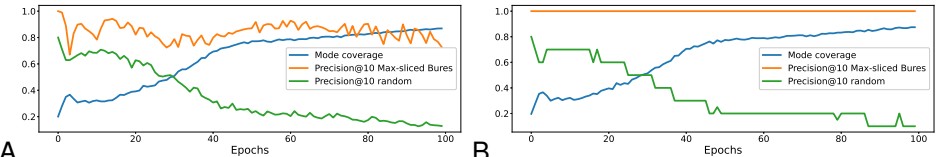

Figure 6: Detecting dropped modes using one-sided max-sliced Bures distance. The slice $\mathbf{w}_{\text{Real}>\text{Fake}}$ is used to identify the top-10 *real* training images with the largest magnitude witness function values at each epoch. Precision@10 measures the fraction that correspond to dropped modes, as compared to random selection. The curves are the mean (A) and median (B) across 100 GAN training trials.

## 4 CONCLUSION

We propose the max-sliced Bures distance, a lower-bound on the max-sliced W2 distance, which can be computed optimally with a tractable algorithm. We show increased performance with kernel-based witness functions for covariate shift detection and correction, and also highlight its utility in the linear case when the feature space is the internal representation of a pre-trained network. Importantly, the one-sided max-sliced Bures divergences enable direct interpretation of under- and over-representation between two samples, which can be used to identify systematic discrepancies.

---

[5]https://github.com/anon-author-dev/gsw

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

## A  APPENDIX

The appendix details the derivation of the proposed divergences, formal results relating them to existing divergences, algorithms, and additional experimental results.

### A.1  DERIVATION OF THE MAX-SLICED KERNEL DIVERGENCES

Let $\mathcal{U}_1 = \{\omega \otimes \omega : |\omega|_2 \leq 1, \omega \in \mathcal{H}\}$ denote the set of rank-1 symmetric operators with bounded trace norm. Let $d$ denote either the TV distance ($d = d_{TV}$) or the squared Bures distance ($d = d_B^2$), defined in equation 4 and equation 5, respectively. In these cases, the expression of the max-sliced distance can be simplified as

$$
\begin{aligned}
max\text{-}d(\rho_X, \rho_Y) &= \sup_{\Omega \in \mathcal{U}_1} d(\Omega \rho_X \Omega, \Omega \rho_Y \Omega) = \sup_{\Omega \in \mathcal{U}_1} d(\langle \Omega, \rho_X \rangle_{HS} \Omega, \langle \Omega, \rho_Y \rangle_{HS} \Omega) \\
&= \sup_{\Omega \in \mathcal{U}_1} \delta(\langle \Omega, \rho_X \rangle_{HS}, \langle \Omega, \rho_Y \rangle_{HS}) = \sup_{\omega \in \mathcal{H}: \|\omega\|_2 \leq 1} \delta(\langle \omega, \rho_X \omega \rangle_{\mathcal{H}}, \langle \omega, \rho_X \omega \rangle_{\mathcal{H}}),
\end{aligned}
$$

where $\langle\cdot,\cdot\rangle_{HS} : (\mathcal{H}\times\mathcal{H})\times(\mathcal{H}\times\mathcal{H})\to\mathbb{R}$ denotes the inner-product defining the Hilbert-Schmidt (Schatten-2 norm), $\langle\Omega,\rho\rangle_{HS} = \langle\omega\otimes\omega,\rho\rangle_{HS} = \langle\omega,\rho\omega\rangle_{\mathcal{H}}$, and $\delta(p,q) = d(p\Omega,q\Omega)$ denotes the distance between scaled versions of $\Omega$, for $p\in\mathbb{R}_{\geq 0}$ and $q\in\mathbb{R}_{\geq 0}$. The equalities follow from the fact that $\Omega\rho\Omega = (\omega\otimes\omega)\,\rho\,(\omega\otimes\omega) = \langle\omega,\rho\omega\rangle\Omega$. The max-sliced TV distance and squared max-sliced Bures distance yield expressions for $\delta(p,q)$ that match the form of the underlying TV and Hellinger divergences: $d_{TV}(p\Omega,q\Omega)$ yields $\delta(p,q) = \frac{1}{2}|p-q|$, and $d_B^2(p\Omega,q\Omega)$ yields $\delta(p,q) = (\sqrt{p}-\sqrt{q})^2$:

$$max\text{-}d_{TV}(\rho_X,\rho_Y) \triangleq \sup_{\omega\in\mathcal{S}}\frac{1}{2}|\langle\omega,\rho_X\omega\rangle - \langle\omega,\rho_Y\omega\rangle|, \quad\text{and} \tag{15}$$

$$max\text{-}d_B(\rho_X,\rho_Y) \triangleq \sup_{\omega\in\mathcal{S}}|\sqrt{\langle\omega,\rho_X\omega\rangle} - \sqrt{\langle\omega,\rho_Y\omega\rangle}|, \tag{16}$$

where $\mathcal{S} = \{\omega\in\mathcal{H} : \|\omega\|_2 = 1\}$. The kernel-based divergences can be expressed in terms of the witness functions since $\langle\omega,\rho_X\omega\rangle = \mathbb{E}\left[\langle\phi(X),\omega\rangle\langle\phi(X),\omega\rangle\right] = \mathbb{E}\left[\omega^2(X)\right]$ and likewise for $\rho_Y$. The underlying divergences and distances are listed in Table 2.

Table 2: Relationship between scalar discrepancy $\delta$, divergence $D$ between continuous $\mu,\nu$ and discrete measures $\boldsymbol{\mu},\boldsymbol{\nu}$, operator dissimilarity $d$, max-sliced dissimilarity, and kernel max-sliced divergence for the TV and squared Bures divergences (squared Bures is twice the squared Hellinger).

| | TV | (Bures)$^2$ |
|---|---|---|
| $\delta(p,q)$ | $\frac{1}{2}|p-q|$ | $|\sqrt{p}-\sqrt{q}|^2$ |
| $D(\boldsymbol{\mu},\boldsymbol{\nu})$ | $\frac{1}{2}\sum_i|\mu_i-\nu_i|$ | $\sum_i\left(\sqrt{\mu_i}-\sqrt{\nu_i}\right)^2$ |
| $D(\mu,\nu)$ | $\frac{1}{2}\int_{\mathcal{X}}|d\mu(x)-d\nu(x)|$ | $\int_{\mathcal{X}}\left(\sqrt{d\mu(x)}-\sqrt{d\nu(x)}\right)^2$ |
| $d(\rho_X,\rho_Y)$ | $\frac{1}{2}\|\rho_X-\rho_Y\|_1$ | $\|\rho_X\|_1 + \|\rho_Y\|_1 - 2\left\|\rho_X^{\frac{1}{2}}\rho_Y^{\frac{1}{2}}\right\|_1$ |
| $max\text{-}d(\rho_X,\rho_Y)$ | $\frac{1}{2}\sup_{\omega\in\mathcal{S}}|\langle\omega,\rho_X\omega\rangle - \langle\omega,\rho_Y\omega\rangle|$ | $\sup_{\omega\in\mathcal{S}}\left(\sqrt{\langle\omega,\rho_X\omega\rangle} - \sqrt{\langle\omega,\rho_Y\omega\rangle}\right)^2$ |
| $max\text{-}D(\mu,\nu)$ | $\frac{1}{2}\sup_{\omega\in\mathcal{S}}|\mathbb{E}\left[\omega^2(X)\right] - \mathbb{E}\left[\omega^2(Y)\right]|$ | $\sup_{\omega\in\mathcal{S}}\left(\sqrt{\mathbb{E}\left[\omega^2(X)\right]} - \sqrt{\mathbb{E}\left[\omega^2(Y)\right]}\right)^2$ |

Slicing is natural for the operator-based distances, as it is inherent in their definition such that they coincide with the corresponding divergences between discrete probability laws (Fuchs & Van De Graaf, 1999). The scalar discrepancy measure $\delta(\cdot,\cdot)$ can be accumulated across a *complete* set of slices to obtain the original distances. Consider a set (or countably infinite sequence) of orthogonal trace-norm operators $\mathcal{O} = \{\Omega_1 = \omega_1\otimes\omega_1, \Omega_2 = \omega_2\otimes\omega_2, \ldots, \Omega_k = \omega_k\otimes\omega_k\}$. Since these operators are orthogonal and have unit trace-norm $\|\sum_{i=1}^k\Omega_i\|_\infty = 1$,

$$d(\rho_X,\rho_Y) \geq \sum_{i=1}^k d(\langle\Omega_i,\rho_X\rangle_{HS}\Omega_i, \langle\Omega_i,\rho_Y\rangle_{HS}\Omega_i) = \sum_{i=1}^k \delta(\langle\Omega_i,\rho_X\rangle_{HS}, \langle\Omega_i,\rho_Y\rangle_{HS})$$

$$= \sum_{i=1}^k \delta(p_i(\mathcal{O}), q_i(\mathcal{O})), \tag{17}$$

where $p_i(\mathcal{O}) = \langle\Omega_i,\rho_X\rangle_{HS}$ and $q_i(\mathcal{O}) = \langle\Omega_i,\rho_Y\rangle_{HS}$. To obtain the equality, one must optimize $\mathcal{O}$ over all possible sets of orthogonal trace-norm operators.

### A.1.1 MAX-SLICED TOTAL VARIATION DISTANCE

The sliced kernel total variation distance is

$$d_{TV}(\Omega\rho_X\Omega, \Omega\rho_Y\Omega) = \frac{1}{2}\|\Omega(\rho_X-\rho_Y)\Omega\|_1 = \frac{1}{2}\|\langle\Omega,\rho_X-\rho_Y\rangle\Omega\|_1 = \frac{1}{2}|\langle\Omega,\rho_X-\rho_Y\rangle|\|\Omega\|_1.$$

Maximizing over slices yields

$$\frac{1}{2}\sup_{\Omega\in\mathcal{U}_1}|\langle\Omega,\rho_X-\rho_Y\rangle|\|\Omega\|_1 = \frac{1}{2}\sup_{\Omega\in\mathcal{U}_1}|\langle\Omega,\rho_X-\rho_Y\rangle| = \frac{1}{2}\sup_{\omega:\|\omega\|_2\leq 1}|\|\omega\|_\mu^2 - \|\omega\|_\nu^2|, \tag{18}$$

where the first equality follows from the fact that distance is maximized when $\|\Omega\|_1 = 1$. This yields the expression

$$max\text{-}D_{TV}^{\mathcal{H}}(\mu, \nu) \triangleq \frac{1}{2} \sup_{\omega : \|\omega\|_2 = 1} |\|\omega\|_\mu^2 - \|\omega\|_\nu^2|. \tag{19}$$

Notably, the penultimate expression in equation 18 can be related to the operator norm,

$$\sup_{\Omega \in \mathcal{U}_1} |\langle \Omega, \rho_X - \rho_Y \rangle| \leq \sup_{\Omega \in \{O \in \mathcal{H} \times \mathcal{H} : \|O\|_1 \leq 1\}} \langle \Omega, \rho_X - \rho_Y \rangle = \|\rho_X - \rho_Y\|_\infty, \tag{20}$$

due to the dual norm definition. Since $\rho_X - \rho_Y$ is symmetric, the equality is achieved. Thus, $max\text{-}D_{TV}^{\mathcal{H}}(\mu, \nu) = \frac{1}{2} \|\rho_X - \rho_Y\|_\infty$. For a linear kernel, this can be computed by finding the largest magnitude eigenvalue of $\boldsymbol{\rho}_X - \boldsymbol{\rho}_Y = \mathbb{E}_{X \sim \mu}[XX^\top] - \mathbb{E}_{Y \sim \nu}[YY^\top] \in \mathbb{R}^{d \times d}$.

### A.1.2 MAX-SLICED BURES DISTANCE

The sliced version of the Bures distance is

$$d_B(\Omega \rho_X \Omega, \Omega \rho_Y \Omega) = \sqrt{\|\Omega \rho_X \Omega\|_1 + \|\Omega \rho_Y \Omega\|_1 - 2\|(\Omega \rho_X \Omega)^{\frac{1}{2}}(\Omega \rho_Y \Omega)^{\frac{1}{2}}\|_1},$$

which can be simplified since $\|\Omega \rho_X \Omega\|_1 = \text{tr}((\omega \otimes \omega)\rho_X(\omega \otimes \omega)) = \langle \omega, \rho_X \omega \rangle \|\omega\|_2^2 = \|\omega\|_\mu^2 \|\omega\|_2^2$, and $\|(\Omega \rho_X \Omega)^{\frac{1}{2}}(\Omega \rho_Y \Omega)^{\frac{1}{2}}\|_1 = \sqrt{\langle \omega, \rho_X \omega \rangle \langle \omega, \rho_Y \omega \rangle} \|\Omega\|_1 = \sqrt{\|\omega\|_\mu^2 \|\omega\|_\nu^2} \|\Omega\|_1 = \|\omega\|_\mu \|\omega\|_\nu \|\omega\|_2^2$. Using these expressions, the max-sliced Bures distance is

$$\sup_{\Omega \in \mathcal{U}_1} d_B(\Omega \rho_X \Omega, \Omega \rho_Y \Omega) = \sup_{\omega \in \mathcal{H} : \|\omega\|_2 \leq 1} \|\omega\|_2 \sqrt{\|\omega\|_\mu^2 + \|\omega\|_\nu^2 - 2\|\omega\|_\mu \|\omega\|_\nu}.$$

The expression is monotonic with the norm of $\omega$ yielding

$$max\text{-}D_B^{\mathcal{H}}(\mu, \nu) \triangleq \sup_{\omega \in \mathcal{H} : \|\omega\|_2 \leq 1} \sqrt{(\|\omega\|_\mu - \|\omega\|_\nu)^2} = \sup_{\omega \in \mathcal{H} : \|\omega\|_2 \leq 1} |\|\omega\|_\mu - \|\omega\|_\nu| \tag{21}$$

$$= \sup_{\omega \in \mathcal{H} : \|\omega\|_2 = 1} \left| \sqrt{\mathbb{E}_{X \sim \mu}[\omega^2(X)]} - \sqrt{\mathbb{E}_{Y \sim \nu}[\omega^2(Y)]} \right|.$$

### A.1.3 MAX-SLICED GAUSS-WASSERSTEIN DISTANCES

Two max-sliced versions of the Gauss-Wasserstein or Fréchet distance in the RKHS are

$$max_L\text{-}D_{GW}^{\mathcal{H}}(\mu, \nu) = \sup_{\Omega \in \mathcal{U}_1} \sqrt{\|\Omega(m_X - m_Y)\|_2^2 + d_B^2(\Omega \Sigma_X \Omega, \Omega \Sigma_Y \Omega)}$$

$$= \sup_{\omega \in \mathcal{H} : \|\omega\|_2 \leq 1} \|\omega\|_2 \sqrt{\langle m_X - m_Y, \omega \rangle^2 + \left(\sqrt{\langle \omega, \Sigma_X \omega \rangle} - \sqrt{\langle \omega, \Sigma_Y \omega \rangle}\right)^2}$$

$$= \sup_{\omega \in \mathcal{H} : \|\omega\|_2 \leq 1} \sqrt{\langle m_X - m_Y, \omega \rangle^2 + \left(\sqrt{\langle \omega, \Sigma_X \omega \rangle} - \sqrt{\langle \omega, \Sigma_Y \omega \rangle}\right)^2}$$

$$= \sup_{\omega \in \mathcal{H} : \|\omega\|_2 \leq 1} \sqrt{(\mathbb{E}_{X \sim \mu}[\omega(X)] - \mathbb{E}_{Y \sim \nu}[\omega(Y)])^2 + \left(\sigma_{\omega(X)} - \sigma_{\omega(Y)}\right)^2}, \tag{22}$$

where $\sigma_{\omega(X)} = \sqrt{\mathbb{E}[(\omega(X) - \langle m_X, \omega \rangle)^2]}$ is the standard deviation of $\omega(X)$, and $\sigma_{\omega(Y)} = \sqrt{\mathbb{E}[(\omega(Y) - \langle m_Y, \omega \rangle)^2]}$, and

$$max_U\text{-}D_{GW}^{\mathcal{H}}(\mu, \nu) = \left( \sup_{\Omega \in \mathcal{U}_1} \|\Omega(m_X - m_Y)\|_2^2 + \sup_{\Omega \in \mathcal{U}_1} d_B^2(\Omega \Sigma_X \Omega, \Omega \Sigma_Y \Omega) \right)^{\frac{1}{2}}$$

$$= \sqrt{MMD^2(\mu, \nu) + \sup_{\omega \in \mathcal{H} : \|\omega\|_2 \leq 1} \left(\sigma_{\omega(X)} - \sigma_{\omega(Y)}\right)^2}$$

$$= \sqrt{MMD^2(\mu, \nu) + max\text{-}d_B^2(\Sigma_X, \Sigma_Y)}. \tag{23}$$

Notably, $max_L\text{-}D_{GW}^{\mathcal{H}}(\mu, \nu)$ is the supremum of the Fréchet distance over all witness functions.

### A.1.4 MAX-SLICED KERNEL WASSERSTEIN

A sliced version of the kernel Wasserstein distance between $\mu$ and $\nu$ relies on the the sliced distance

$$
\begin{aligned}
d_{\kappa,\omega}(X,Y) &= \|(\omega \otimes \omega)(\phi(X) - \phi(Y))\|_2 = \|\langle \omega, \phi(X) - \phi(Y)\rangle \omega\|_2 \\
&= |\langle \omega, \phi(X) - \phi(Y)\rangle|\|\omega\|_2 = |\langle \omega, \phi(X)\rangle - \langle \omega, \phi(Y)\rangle|\|\omega\|_2 = |\omega(X) - \omega(Y)|\|\omega\|_2.
\end{aligned}
$$

This distance is monotonic with the norm of $\omega$ and convex with respect to $\omega$. The max-sliced kernel Wasserstein-$p$ distance, $p \geq 1$, is

$$
\begin{aligned}
\textit{max-}W_p^{\mathcal{H}}(\mu,\nu) &= \sup_{\omega \in \mathcal{H}:\|\omega\|_2 \leq 1} \inf_{\gamma \in \Gamma(\mu,\nu)} \left[ \mathop{\mathbb{E}}_{(X,Y)\sim\gamma} (d_{\kappa,\omega}(X,Y))^p \right]^{\frac{1}{p}} \\
&= \sup_{\omega \in \mathcal{H}:\|\omega\|_2 \leq 1} \inf_{\gamma \in \Gamma(\mu,\nu)} \left[ \mathop{\mathbb{E}}_{(X,Y)\sim\gamma} |\omega(X) - \omega(Y)|^p \right]^{\frac{1}{p}},
\end{aligned}
\tag{24}
$$

which is a one-dimensional optimal transport problem. Let $\omega_{\#}\mu$ and $\omega_{\#}\nu$ denote the pushforward measures, then the divergence can be written as

$$
\textit{max-}W_p^{\mathcal{H}}(\mu,\nu) = \sup_{\omega \in \mathcal{H}:\|\omega\|_2 \leq 1} \inf_{\pi \in \Pi(\omega_{\#}\mu,\omega_{\#}\nu)} \left[ \mathop{\mathbb{E}}_{(S,T)\sim\pi} |S - T|^p \right]^{\frac{1}{p}},
\tag{25}
$$

where $\Pi(\omega_{\#}\mu, \omega_{\#}\nu)$ is the set of all joint distributions coupling the pushforward measures.

Assuming the measures are absolutely continuous and adopting the notation from Santambrogio (2015), let $F_{\omega,\mu}(w) = \int_{-\infty}^{w} d\omega_{\#}\mu = \omega_{\#}\mu((-\infty, w])$ and $F_{\omega,\nu}(w) = \int_{-\infty}^{w} d\omega_{\#}\nu = \omega_{\#}\nu((-\infty, w])$ denote the cumulative distribution functions of the pushforward measures with pseudo-inverses $F_{\omega,\mu}^{-1}(q) = \inf\{w \in \mathbb{R} : F_{\omega,\mu}(w) \geq q\}$ and $F_{\omega,\nu}^{-1}(q) = \inf\{w \in \mathbb{R} : F_{\omega,\nu}(w) \geq q\}$. As shown in Lemma 2.8 (Santambrogio, 2015), then the optimal transport plan $\pi^*$ has cumulative distribution $G_\omega(w_X, w_Y) = \min\{F_{\omega,\mu}(w_X), F_{\omega,\nu}(w_Y)\}$ and the divergence is

$$
\textit{max-}W_p^{\mathcal{H}}(\mu,\nu) = \sup_{\omega \in \mathcal{H}:\|\omega\|_2 \leq 1} \left[ \int_0^1 \left| F_{\omega,\mu}^{-1}(q) - F_{\omega,\nu}^{-1}(q) \right|^p dq \right]^{\frac{1}{p}},
\tag{26}
$$

and for the case $p = 1$ (Santambrogio, 2015, Proposition 2.17),

$$
\textit{max-}W_1^{\mathcal{H}}(\mu,\nu) \triangleq \sup_{\omega \in \mathcal{H}:\|\omega\|_2 \leq 1} \int_{\mathbb{R}} |F_{\omega,\mu}(w) - F_{\omega,\nu}(w)|\, dw.
\tag{27}
$$

The objective in the last quantity is an $L_1$-norm version of the Cramér–von Mises criterion (Schmid & Trede, 1995; Anderson, 1962). That is, the max-sliced kernel Wasserstein-1 distance is equivalent to a max-sliced $L_1$-norm version of the Cramér–von Mises criterion, where the slicing corresponds to a function in the RKHS that witnesses the largest discrepancies between the measures. The choice of $p = 2$ simplifies, yielding the following optimization problem

$$
\begin{aligned}
\textit{max-}W_2^{\mathcal{H}}(\mu,\nu) &= \sup_{\omega \in \mathcal{H}:\|\omega\|_2 \leq 1} \inf_{\gamma \in \Gamma(\mu,\nu)} \left( \mathop{\mathbb{E}}_{(X,Y)\sim\gamma} \left[ \omega^2(X) + \omega^2(Y) - 2\omega(X)\omega(Y) \right] \right)^{\frac{1}{2}} \\
&= \sup_{\omega \in \mathcal{H}:\|\omega\|_2 \leq 1} \left( \|\omega\|_\mu^2 + \|\omega\|_\nu^2 - \sup_{\gamma \in \Gamma(\mu,\nu)} \mathop{\mathbb{E}}_{(X,Y)\sim\gamma} \left[ 2\omega(X)\omega(Y) \right] \right)^{\frac{1}{2}} \\
&= \sup_{\omega \in \mathcal{H}:\|\omega\|_2 \leq 1} \left( \|\omega\|_\mu^2 + \|\omega\|_\nu^2 - 2 \int_0^1 F_{\omega,\mu}^{-1}(q) F_{\omega,\nu}^{-1}(q) dq \right)^{\frac{1}{2}}.
\end{aligned}
\tag{28}
\tag{29}
$$

### A.2 RELATIONSHIP BETWEEN THE KERNEL-BASED MAX-SLICED DIVERGENCES

The Bures distance can be used to lower bound the TV distance, $d_B^2(\rho_X, \rho_Y) \leq \|\rho_X - \rho_Y\|_1 = 2d_{TV}(\rho_X, \rho_Y)$ (Fuchs & Van De Graaf, 1999). This inequality stems from the inequality between the squared Hellinger and total variation distances for discrete probability laws. $\sum_i \|\sqrt{p_i} - \sqrt{q_i}\|_2^2 \leq \sum_i |p_i - q_i|$, where the inequality holds for each summand, $|\sqrt{p_i} - \sqrt{q_i}|^2 \leq |\sqrt{p_i} - \sqrt{q_i}|(\sqrt{p_i} + \sqrt{q_i}) = |p_i - q_i|$. This inequality holds for the max-sliced divergences as stated in Theorem 1.

*Proof of Theorem 1.* $\left(\|\omega\|_\mu - \|\omega\|_\nu\right)^2 \leq \left|\|\omega\|_\mu^2 - \|\omega\|_\nu^2\right|$, where the inequality is a consequence of the inequality between the arithmetic and geometric means, let $a = \|\omega\|_\mu^2$ and $b = \|\omega\|_\nu^2$, then $(\sqrt{a} - \sqrt{b})^2 \leq |\sqrt{a} - \sqrt{b}|(\sqrt{a} + \sqrt{b}) = |a - b|$. Taking the supremum over $\omega \in \mathcal{H}$ with the constraint $\|\omega\|_2 \leq 1$ yields the desired the result $\left(max\text{-}D_B^{\mathcal{H}}(\mu, \nu)\right)^2 \leq 2\left(max\text{-}D_{TV}^{\mathcal{H}}(\mu, \nu)\right)$.

$\square$

We now consider the relationship between the Gauss-Wasserstein or Fréchet distance—which combines the distances between the first and second-order moments—with the max-sliced Bures when it is applied directly to the uncentered covariance matrices. For this we need the following lemma.

**Lemma 4** (Reverse triangle inequality). *For two vectors in $\mathbb{R}^2$, the difference between their Euclidean norms is less than or equal to the Euclidean norm of their differences. For $a, b, c, d \in \mathbb{R}$, $\left|\sqrt{a^2 + b^2} - \sqrt{c^2 + d^2}\right| \leq \sqrt{(a - c)^2 + (b - d)^2}$.*

*Proof.* Let $e = \left(\sqrt{a^2 + b^2} - \sqrt{c^2 + d^2}\right)^2$ and $f = (a - c)^2 + (b - d)^2$.

$e = a^2 + b^2 + c^2 + d^2 - 2\sqrt{(a^2 + b^2)(c^2 + d^2)}$

$\quad = a^2 + b^2 + c^2 + d^2 - 2\sqrt{a^2c^2 + b^2d^2 + b^2c^2 + a^2d^2}$

$\quad \leq a^2 + b^2 + c^2 + d^2 - 2\sqrt{a^2c^2 + b^2d^2 + 2\sqrt{a^2b^2c^2d^2}}$, by the arithmetic and geometric mean inequality

$\quad = a^2 + b^2 + c^2 + d^2 - 2\sqrt{(\sqrt{a^2c^2} + \sqrt{b^2d^2})^2}$

$\quad = a^2 + b^2 + c^2 + d^2 - 2(\sqrt{a^2c^2} + \sqrt{b^2d^2})$

$\quad \leq a^2 + b^2 + c^2 + d^2 - 2(ac + bd) = (a - c)^2 + (b - d)^2 = f.$

Taking the square root of each side yields the inequality $\sqrt{e} \leq \sqrt{f}$. $\square$

*Proof of Theorem 2.* To relate the sliced Bures and Gauss-Wasserstein distances, we note that $\|\omega\|_\mu^2 = \langle \rho_X, \omega \otimes \omega \rangle_{HS} = \langle \Sigma_X + m_X \otimes m_X, \omega \otimes \omega \rangle_{HS} = \sigma_{\omega(X)}^2 + \langle m_X, \omega \rangle^2$ and likewise for $\|\omega\|_\nu^2$. Then,

$$\left|\sqrt{\|\omega\|_\mu^2} - \sqrt{\|\omega\|_\nu^2}\right| = \left|\sqrt{\langle m_X, \omega \rangle^2 + \sigma_{\omega(X)}^2} - \sqrt{\langle m_Y, \omega \rangle^2 + \sigma_{\omega(Y)}^2}\right|$$

$$\leq \sqrt{\langle m_X - m_Y, \omega \rangle^2 + (\sigma_{\omega(X)} - \sigma_{\omega(Y)})^2}, \quad (30)$$

where the inequality relies on Lemma 4, with $a = \langle m_X, \omega \rangle$, $b = \sigma_{\omega(X)}$, $c = \langle m_Y, \omega \rangle$, and $d = \sigma_{\omega(Y)}$. Taking supremum over slices yields the desired inequality. $\square$

Theorem 2 shows that the max-sliced kernel Bures distance is a lower-bound on the max-sliced kernel Wasserstein-2 distance, since the latter is lower bounded by the max-sliced kernel Gauss-Wasserstein distance.

*Proof of Theorem 3.* Let $\omega(X) = \langle m_X, \omega \rangle + \tilde{\omega}(X)$ and $\omega(Y) = \langle m_Y, \omega \rangle + \tilde{\omega}(Y)$, where $\tilde{\omega}(X) = \langle \phi(X) - m_X, \omega \rangle$ and $\tilde{\omega}(Y) = \langle \phi(Y) - m_Y, \omega \rangle$ are zero mean, and $\mathbb{E}[\tilde{\omega}^2(X)] = \sigma_{\omega(X)}^2$ and $\mathbb{E}[\tilde{\omega}^2(Y)] = \sigma_{\omega(Y)}^2$. The squared Fréchet distance between random variables $\omega(X)$ and $\omega(Y)$ is

$$(\mathbb{E}[\omega(X)] - \mathbb{E}[\omega(Y)])^2 + (\sigma_{\omega(X)} - \sigma_{\omega(Y)})^2 = \langle m_X - m_Y, \omega \rangle^2 + (\sigma_{\omega(X)} - \sigma_{\omega(Y)})^2$$

$$= \langle m_X, \omega \rangle^2 + \sigma_{\omega(X)}^2 + \langle m_Y, \omega \rangle^2 + \sigma_{\omega(Y)}^2 - 2\left(\langle m_X, \omega \rangle \langle m_Y, \omega \rangle + \sigma_{\omega(X)}\sigma_{\omega(Y)}\right)$$

$$= \mathbb{E}[\omega^2(X)] + \mathbb{E}[\omega^2(Y)] - 2\left(\mathbb{E}[\omega(X)]\mathbb{E}[\omega(Y)] + \sqrt{\mathbb{E}[\tilde{\omega}^2(X)]\mathbb{E}[\tilde{\omega}^2(Y)]}\right).$$

By Hölder's inequality, $\mathbb{E}[\tilde{\omega}(X)\tilde{\omega}(Y)] \leq \sigma_{\omega(X)}\sigma_{\omega(Y)} = \sqrt{\mathbb{E}[\tilde{\omega}^2(X)]\mathbb{E}[\tilde{\omega}^2(Y)]}$. Consequently,

$$\mathbb{E}[\omega^2(X)] + \mathbb{E}[\omega^2(Y)] - 2\left(\mathbb{E}[\omega(X)]\mathbb{E}[\omega(Y)] + \sqrt{\mathbb{E}[\tilde{\omega}^2(X)]\mathbb{E}[\tilde{\omega}^2(Y)]}\right) \quad (31)$$

$$\leq \mathbb{E}[\omega^2(X)] + \mathbb{E}[\omega^2(Y)] - 2(\mathbb{E}[\omega(X)]\mathbb{E}[\omega(Y)] + \mathbb{E}[\tilde{\omega}(X)\tilde{\omega}(Y)])$$

$$= \mathbb{E}[\omega^2(X)] + \mathbb{E}[\omega^2(Y)] - 2\mathbb{E}[\omega(X)\omega(Y)] = \mathbb{E}[(\omega(X) - \omega(Y))^2]. \quad (32)$$

Taking the infimum over all possible joint distributions $(X, Y) \sim \gamma$ that are within the coupling distribution $\gamma \in \Gamma$, yields the sliced Wasserstein-2 distance on the right hand side. Maximizing over slices, yields $max_U\text{-}D_{GW}^{\mathcal{H}}(\mu, \nu) \leq max\text{-}W_2^{\mathcal{H}}(\mu, \nu)$ and combining with Theorem 2 yields the desired result. $\qquad\square$

### A.3 RELATIONSHIP TO OTHER DIVERGENCES

Finally, we note that the terms in the max-sliced Gauss-Wasserstein divergences are related to kernel Fischer discriminant analysis (KFDA) (Mika et al., 1999). KFDA objective is based on the ratio of the difference in means to the pooled variances:

$$D_{FDA}^{\mathcal{H}}(\mu, \nu) = \sup_\omega \frac{\langle \omega, m_X - m_Y \rangle^2}{\sigma_{\omega(X)}^2 + \sigma_{\omega(Y)}^2} = \sup_\omega \frac{\langle \omega, m_X - m_Y \rangle^2}{\langle \omega, (\Sigma_X + \Sigma_Y)\omega \rangle}. \tag{33}$$

KFDA seeks a witness function which has widely separated means for the two measures, and minimal variance.

### A.4 COMPUTING THE MAX-SLICED KERNEL DIVERGENCES

We assume the witness function[6] $\omega \in \mathcal{H}$ is of the form $\omega = \sum_{i=1}^{n+m} \alpha_i \phi(z_i)$ with $\omega(\cdot) = \sum_{i=1}^{n+m} \alpha_i \kappa(\cdot, z_i)$ where $z_i = \begin{cases} x_i, & 1 \leq i \leq m \\ y_{i-m}, & m < i \leq n+m \end{cases}$ and $\boldsymbol{\alpha} \in \mathbb{R}^{m+n}$. In this case,

$$\|\omega\|_{\hat{\mu}}^2 = \langle \omega \otimes \omega, (\phi \otimes \phi)_\# \hat{\mu} \rangle = \sum_{j=1}^m \mu_j \langle \omega \otimes \omega, \phi(x_j) \otimes \phi(x_j) \rangle = \sum_{j=1}^m \mu_j \langle \omega, \phi(x_j) \rangle^2$$

$$= \sum_{j=1}^m \mu_j \langle \sum_{i=1}^{n+m} \alpha_i \phi(z_i), \phi(x_j) \rangle^2 = \sum_{j=1}^m \mu_j \left( \sum_{i=1}^{n+m} \alpha_i \kappa(z_i, x_j) \right)^2 = \langle \boldsymbol{\mu}, (\mathbf{K}_{XZ}\boldsymbol{\alpha})^{\circ 2} \rangle$$

$$= \boldsymbol{\alpha}^\top \mathbf{K}_{XZ}^\top \text{diag}(\boldsymbol{\mu}) \mathbf{K}_{XZ}\boldsymbol{\alpha} = \|\mathbf{D}_{\boldsymbol{\mu}}^{\frac{1}{2}} \mathbf{K}_{XZ}\boldsymbol{\alpha}\|_2^2,$$

where $\kappa(z_i, z_j) = K_{ij}$, $\mathbf{K} = \begin{bmatrix} \mathbf{K}_{XX}, \mathbf{K}_{XY} \\ \mathbf{K}_{YX}, \mathbf{K}_{YY} \end{bmatrix} = \begin{bmatrix} \mathbf{K}_{XZ} \\ \mathbf{K}_{YZ} \end{bmatrix}$, $(\cdot)^{\circ 2}$ denotes the elementwise squaring of a matrix/vector, and $\mathbf{D_v} = \text{diag}(\mathbf{v})$ denotes a diagonal matrix whose diagonal entries are the vector $\mathbf{v}$. Similarly, $\|\omega\|_{\hat{\nu}}^2 = \langle \boldsymbol{\nu}, (\mathbf{K}_{YZ}\boldsymbol{\alpha})^{\circ 2} \rangle = \|\mathbf{D}_{\boldsymbol{\nu}}^{\frac{1}{2}} \mathbf{K}_{YZ}\boldsymbol{\alpha}\|_2^2$.

In order for the constraint $\|\omega\|_2^2 \leq 1 \implies \boldsymbol{\alpha}^\top \mathbf{K}\boldsymbol{\alpha} \leq 1$ to ensure a bounded solution, we assume $\mathbf{K}$ is strictly positive definite. For this purpose, we add a small value to its diagonal $\mathbf{K} + 10^{-9}\mathbf{I}$ when necessary in the optimization procedures. For computational purposes when $m$ or $n$ are large, a subset (possibly random) of landmark points of size $l < n + m$ can be used to form the witness function $\omega = \sum_{i=1}^l \alpha_i \phi(z_{\tau_i})$, where $\{\tau_i\}_{i=1}^l \subset \{1, \ldots, m+n\}$. In this case, $\mathbf{K}_{XZ} \in \mathbb{R}^{m \times l}$ and $\mathbf{K}_{YZ} \in \mathbb{R}^{n \times l}$ with $[\mathbf{K}_{XZ}]_{i,j} = \kappa(x_i, z_{\tau_j})$ and $[\mathbf{K}_{YZ}]_{i,j} = \kappa(y_i, z_{\tau_j})$. In this case, the constraint also needs to be adjusted.

#### A.4.1 MAX-SLICED KERNEL TV DISTANCE

Using these expressions, the max-sliced kernel TV distance is

$$max\text{-}D_{TV}^{\mathcal{H}}(\hat{\mu}, \hat{\nu}) = \max_{\boldsymbol{\alpha}:\boldsymbol{\alpha}^\top \mathbf{K}\boldsymbol{\alpha} \leq 1} \frac{1}{2} \left| \|\mathbf{D}_{\boldsymbol{\mu}}^{\frac{1}{2}} \mathbf{K}_{XZ}\boldsymbol{\alpha}\|_2^2 - \|\mathbf{D}_{\boldsymbol{\nu}}^{\frac{1}{2}} \mathbf{K}_{YZ}\boldsymbol{\alpha}\|_2^2 \right| \tag{34}$$

$$= \max_{\boldsymbol{\alpha}:\boldsymbol{\alpha}^\top \mathbf{K}\boldsymbol{\alpha} \leq 1} \frac{1}{2} |\boldsymbol{\alpha}^\top (\mathbf{K}_{XZ}^\top \mathbf{D}_{\boldsymbol{\mu}} \mathbf{K}_{XZ} - \mathbf{K}_{YZ}^\top \mathbf{D}_{\boldsymbol{\nu}} \mathbf{K}_{YZ})\boldsymbol{\alpha}|. \tag{35}$$

The solution is the generalized eigenvector corresponding to the largest magnitude eigenvalue of the generalized eigenvalue problem $\mathbf{A}\mathbf{v} = \lambda \mathbf{K}\mathbf{v}$, where $\mathbf{A} = \mathbf{K}_{XZ}^\top \mathbf{D}_{\boldsymbol{\mu}} \mathbf{K}_{XZ} - \mathbf{K}_{YZ}^\top \mathbf{D}_{\boldsymbol{\nu}} \mathbf{K}_{YZ}$. $\boldsymbol{\alpha}^\star = \arg\max_{\boldsymbol{\alpha}} \frac{\boldsymbol{\alpha}^\top \mathbf{A}\boldsymbol{\alpha}}{\boldsymbol{\alpha}^\top \mathbf{K}\boldsymbol{\alpha}}$. The witness function is $\omega^\star(\cdot) = \sum_{i=1}^{m+n} \alpha_i^\star \kappa(\cdot, z_i)$.

---

[6]Similar to kernel PCA, the constraint $\|\omega\|_2 \leq 1$ allows the use of the representer theorem for the RKHS.

### A.4.2 MAX-SLICED KERNEL BURES DISTANCE

The sample-based max-sliced kernel Bures distance is

$$\text{max-}D_B^{\mathcal{H}}(\hat{\mu}, \hat{\nu}) = \max_{\boldsymbol{\alpha}:\boldsymbol{\alpha}^\top \mathbf{K}\boldsymbol{\alpha} \leq 1} \left| \|\mathbf{D}_{\boldsymbol{\mu}}^{\frac{1}{2}} \mathbf{K}_{XZ} \boldsymbol{\alpha}\|_2 - \|\mathbf{D}_{\boldsymbol{\nu}}^{\frac{1}{2}} \mathbf{K}_{YZ} \boldsymbol{\alpha}\|_2 \right| \tag{36}$$

$$= \max_{s \in \{-1, +1\}} \max_{\boldsymbol{\alpha}:\boldsymbol{\alpha}^\top \mathbf{K}\boldsymbol{\alpha} \leq 1} s\|\mathbf{D}_{\boldsymbol{\mu}}^{\frac{1}{2}} \mathbf{K}_{XZ} \boldsymbol{\alpha}\|_2 - s\|\mathbf{D}_{\boldsymbol{\nu}}^{\frac{1}{2}} \mathbf{K}_{YZ} \boldsymbol{\alpha}\|_2 \tag{37}$$

The last expression shows that the max-sliced Bures distance can be expressed as a bilevel optimization problem, where the inner optimization problem—which we refer to as one-sided max-sliced Bures divergence—is a difference of convex functions:

$$\text{max-}D_B^{\mathcal{H}}(\hat{\mu}, \hat{\nu}) = \max \left\{ \left[ \min_{\boldsymbol{\alpha}:\boldsymbol{\alpha}^\top \mathbf{K}\boldsymbol{\alpha} \leq 1} g(\boldsymbol{\alpha}) - h(\boldsymbol{\alpha}) \right], \left[ \min_{\boldsymbol{\alpha}:\boldsymbol{\alpha}^\top \mathbf{K}\boldsymbol{\alpha} \leq 1} h(\boldsymbol{\alpha}) - g(\boldsymbol{\alpha}) \right] \right\}, \tag{38}$$

$$g(\boldsymbol{\alpha}) = \|\mathbf{D}_{\boldsymbol{\nu}}^{\frac{1}{2}} \mathbf{K}_{YZ} \boldsymbol{\alpha}\|_2, \tag{39}$$

$$h(\boldsymbol{\alpha}) = \|\mathbf{D}_{\boldsymbol{\mu}}^{\frac{1}{2}} \mathbf{K}_{XZ} \boldsymbol{\alpha}\|_2. \tag{40}$$

Without loss of generality, we will consider the first case ($s = 1$),

$$\min_{\boldsymbol{\alpha}:\boldsymbol{\alpha}^\top \mathbf{K}\boldsymbol{\alpha} \leq 1} g(\boldsymbol{\alpha}) - h(\boldsymbol{\alpha}). \quad \text{(P)}$$

Inspired by the approach Landsman (2008), we relate this problem to a quadratic program, specifically, the quadratically constrained quadratic program

$$\min_{\boldsymbol{\alpha}:\boldsymbol{\alpha}^\top \mathbf{K}\boldsymbol{\alpha} \leq 1} c_1 g^2(\boldsymbol{\alpha}) - c_2 h^2(\boldsymbol{\alpha}) = \min_{\boldsymbol{\alpha}:\boldsymbol{\alpha}^\top \mathbf{K}\boldsymbol{\alpha} \leq 1} \boldsymbol{\alpha}^\top (c_1 \mathbf{K}_{YZ}^\top \mathbf{D}_{\boldsymbol{\nu}} \mathbf{K}_{YZ} - c_2 \mathbf{K}_{XZ}^\top \mathbf{D}_{\boldsymbol{\mu}} \mathbf{K}_{XZ}) \boldsymbol{\alpha}, \quad \text{(Q)},$$

for $c_1, c_2 \in \mathbb{R}_{\geq 0}$. The solution of which can be obtained as in the max-sliced kernel TV distance by solving a generalized eigenvalue problem. For (P), the Lagrangian function is

$$L(\boldsymbol{\alpha}, \lambda) = g(\boldsymbol{\alpha}) - h(\boldsymbol{\alpha}) - \lambda(\boldsymbol{\alpha}^\top \mathbf{K}\boldsymbol{\alpha} - 1). \tag{41}$$

Let $\mathcal{G} = \{\boldsymbol{\alpha} : g(\boldsymbol{\alpha}) > 0, h(\boldsymbol{\alpha}) > 0\}$ denote the set of points where $g$ and $h$ are differentiable. Then for $\boldsymbol{\alpha} \in \mathcal{G}$ and $\mathbf{K}$ positive definite, $L(\boldsymbol{\alpha}, \lambda)$ is differentiable, and

$$\nabla_{\boldsymbol{\alpha}} L(\boldsymbol{\alpha}, \lambda) = \nabla_{\boldsymbol{\alpha}} g(\boldsymbol{\alpha}) - \nabla_{\boldsymbol{\alpha}} h(\boldsymbol{\alpha}) - 2\lambda \mathbf{K}\boldsymbol{\alpha} = \frac{1}{2g(\boldsymbol{\alpha})} \nabla_{\boldsymbol{\alpha}} g^2(\boldsymbol{\alpha}) - \frac{1}{2h(\boldsymbol{\alpha})} \nabla_{\boldsymbol{\alpha}} h^2(\boldsymbol{\alpha}) - 2\lambda \mathbf{K}\boldsymbol{\alpha},$$

where the equality follows from $\nabla_{\boldsymbol{\alpha}} g^2(\boldsymbol{\alpha}) = 2g(\boldsymbol{\alpha}) \nabla_{\boldsymbol{\alpha}} g(\boldsymbol{\alpha})$ and likewise $\nabla_{\boldsymbol{\alpha}} h^2(\boldsymbol{\alpha}) = 2h(\boldsymbol{\alpha}) \nabla_{\boldsymbol{\alpha}} h(\boldsymbol{\alpha})$. For (Q), the Lagrangian function and its gradient are

$$\bar{L}(\boldsymbol{\alpha}, \lambda) = c_1 g^2(\boldsymbol{\alpha}) - c_2 h^2(\boldsymbol{\alpha}) - \lambda(\boldsymbol{\alpha}^\top \mathbf{K}\boldsymbol{\alpha} - 1), \tag{42}$$

$$\nabla_{\boldsymbol{\alpha}} \bar{L}(\boldsymbol{\alpha}, \lambda) = c_1 \nabla_{\boldsymbol{\alpha}} g^2(\boldsymbol{\alpha}) - c_2 \nabla_{\boldsymbol{\alpha}} h^2(\boldsymbol{\alpha}) - 2\lambda \mathbf{K}\boldsymbol{\alpha}. \tag{43}$$

If $c_1 = \frac{1}{2g(\boldsymbol{\alpha})}$ and $c_2 = \frac{1}{2h(\boldsymbol{\alpha})}$, then $\nabla_{\boldsymbol{\alpha}} L(\boldsymbol{\alpha}, \lambda) = \nabla_{\boldsymbol{\alpha}} \bar{L}(\boldsymbol{\alpha}, \lambda)$.

Let $\boldsymbol{\alpha}^*$ denote a global optimum of (Q). If $c_1 = \frac{1}{2g(\boldsymbol{\alpha}^*)}$ and $c_2 = \frac{1}{2h(\boldsymbol{\alpha}^*)}$, then $\nabla_{\boldsymbol{\alpha}} L(\boldsymbol{\alpha}, \lambda)|_{\boldsymbol{\alpha}=\boldsymbol{\alpha}^*} = \nabla_{\boldsymbol{\alpha}} \bar{L}(\boldsymbol{\alpha}, \lambda)|_{\boldsymbol{\alpha}=\boldsymbol{\alpha}^*}$. Consequently, $\boldsymbol{\alpha}^*$ is a local optimum of (P). By the Karush–Kuhn–Tucker conditions, it is a necessary condition for all optima of (P) in $\mathcal{G}$ to have this form. Thus, any global optimum of (P) that lies in $\mathcal{G}$ corresponds to a global optimum of (Q) for particular values of $c_1, c_2$. The family of solutions to (Q) that includes **all** local optima of (P) in $\mathcal{G}$, is

$$\boldsymbol{\alpha}_\gamma^* = \arg\max_{\boldsymbol{\alpha}:\boldsymbol{\alpha}^\top \mathbf{K}\boldsymbol{\alpha} \leq 1} \gamma h^2(\boldsymbol{\alpha}) - g^2(\boldsymbol{\alpha}), \quad \gamma = \frac{c_2}{c_1} \in (0, 1], \tag{44}$$

where the bounds are due to the non-negative functions and $h^2(\boldsymbol{\alpha}_\gamma^*) \geq g^2(\boldsymbol{\alpha}_\gamma^*) \implies \frac{1}{2c_2} \geq \frac{1}{2c_1} \implies c_1 \geq c_2 \implies \frac{c_2}{c_1} \leq 1$. Notably, $\gamma = 1 \implies c_1 = c_2$ corresponds to the one-sided max-sliced TV. The global optimum of (P) within $\mathcal{G}$ is necessarily within $\{\boldsymbol{\alpha}_\gamma^*\}_{\gamma \in (0,1]}$ and can be found as the solution to the bound scalar optimization problem $\min_{\gamma \in (0,1]} g(\boldsymbol{\alpha}_\gamma^*) - h(\boldsymbol{\alpha}_\gamma^*)$.

A remaining case for a global optimum is a non-differentiable point $\boldsymbol{\alpha}^\star \notin \mathcal{G}$, specifically, $g(\boldsymbol{\alpha}^\star) = 0$ (the case of $h(\boldsymbol{\alpha}^\star) = 0$ is trivial), which corresponds to $\boldsymbol{\alpha}^\star \in$

$\text{Null}(\mathbf{D}_{\boldsymbol{\nu}}\mathbf{K}_{YZ})$. In this case, the generalized eigenvalue problem must be restricted to the nullspace $\boldsymbol{\alpha}^*_\varnothing = \arg\max_{\boldsymbol{\alpha}\in\text{Null}(\mathbf{D}_{\boldsymbol{\nu}}\mathbf{K}_{YZ}):\boldsymbol{\alpha}^\top\mathbf{K}\boldsymbol{\alpha}\leq 1} h^2(\boldsymbol{\alpha})$. Let $\mathbf{V} \in \mathbb{R}^{(m+n)\times p}$ denote a matrix of $p$ orthonormal columns that spans the nullspace, then $\boldsymbol{\alpha}^*_\varnothing = \mathbf{V}\boldsymbol{\beta}^*$, where $\boldsymbol{\beta}^* = \arg\max_{\boldsymbol{\beta}\in\mathbb{R}^p:\boldsymbol{\beta}^\top\mathbf{V}^\top\mathbf{K}\mathbf{V}\boldsymbol{\beta}\leq 1}\|\mathbf{D}_{\boldsymbol{\mu}}^{\frac{1}{2}}\mathbf{K}_{XZ}\mathbf{V}\boldsymbol{\beta}\|_2^2$. Overall, the one-sided max-sliced kernel Bures distance can be computed using a combination of a line search for $\gamma \in (0,1]$ and checking the solution in the nullspace as described in Algorithm 1.

---

**Algorithm 1:** One-sided max-sliced kernel Bures divergence

---

**Input:** $\{(x_i,\mu_i)\}_{i=1}^m, \{(y_i,\nu_i)\}_{i=1}^n, \kappa(\cdot,\cdot), \tau \subseteq \{1, m+n\}$

1   $z_i = \begin{cases} x_i & i \leq m, \\ y_{i-m} & i+1 \leq i \leq m+n \end{cases}$

2   $\mathbf{D}_{\boldsymbol{\mu}}^{\frac{1}{2}}\mathbf{K}_{XZ} = [\sqrt{\mu_i}\kappa(x_i, z_{\tau_j})]_{i=1,j=1}^{m,\,|\tau|}$

3   $\mathbf{D}_{\boldsymbol{\nu}}^{\frac{1}{2}}\mathbf{K}_{YZ} = [\sqrt{\nu_i}\kappa(y_i, z_{\tau_j})]_{i=1,j=1}^{n,\,|\tau|}$

4   $\mathbf{A} = \mathbf{K}_{XZ}^\top\mathbf{D}_{\boldsymbol{\mu}}\mathbf{K}_{XZ}$

5   $\mathbf{B} = \mathbf{K}_{YZ}^\top\mathbf{D}_{\boldsymbol{\nu}}\mathbf{K}_{YZ}$

6   $\mathbf{K} = [\kappa(z_{\tau_i}, z_{\tau_j})]_{i=1,j=1}^{m,\,|\tau|}$

7   $\boldsymbol{\alpha}^\star = \text{ONESIDEDMAXSLICEDKERNELBURES}(\mathbf{A}, \mathbf{B}, \mathbf{K})$

8   $\omega_{\mu>\nu}(z): z \mapsto \sum_{i=1}^{|\tau|}\alpha_i^\star\kappa(z, z_{\tau_i})$
    **Output:** $\omega_{\mu>\nu}$

---

1   **ONESIDEDMAXSLICEDKERNELBURES** (A,B,K)

2     $\boldsymbol{\alpha}_\gamma: \gamma \mapsto \arg\max_{\boldsymbol{\alpha}:\boldsymbol{\alpha}^\top\mathbf{K}\boldsymbol{\alpha}\leq 1}\boldsymbol{\alpha}^\top(\gamma\mathbf{A}-\mathbf{B})\boldsymbol{\alpha}$

3     $\gamma^\star = \arg\max_{0<\gamma\leq 1}\sqrt{\boldsymbol{\alpha}_\gamma^\top\mathbf{A}\boldsymbol{\alpha}_\gamma} - \sqrt{\boldsymbol{\alpha}_\gamma^\top\mathbf{B}\boldsymbol{\alpha}_\gamma}$

4     $\boldsymbol{\alpha}_\varnothing = \arg\max_{\boldsymbol{\alpha}\in\text{Null}(\mathbf{B}):\boldsymbol{\alpha}^\top\mathbf{K}\boldsymbol{\alpha}\leq 1}\boldsymbol{\alpha}^\top\mathbf{A}\boldsymbol{\alpha}$

5     **if** $\sqrt{\boldsymbol{\alpha}_\varnothing^\top\mathbf{A}\boldsymbol{\alpha}_\varnothing} > \sqrt{\boldsymbol{\alpha}_{\gamma^\star}^\top\mathbf{A}\boldsymbol{\alpha}_{\gamma^\star}} - \sqrt{\boldsymbol{\alpha}_{\gamma^\star}^\top\mathbf{B}\boldsymbol{\alpha}_{\gamma^\star}}$ **then**

6       $\boldsymbol{\alpha}^\star = \boldsymbol{\alpha}_\varnothing$

7     **else**

8       $\boldsymbol{\alpha}^\star = \boldsymbol{\alpha}_{\gamma^\star}$

9     **end**

10    **return** $\boldsymbol{\alpha}^\star$

---

### A.4.3 MAX-SLICED KERNEL WASSERSTEIN

The empirical version of max-sliced kernel Wasserstein divergence can be expressed in terms of either equation 24 or equation 26,

$$max\text{-}W_p^{\mathcal{H}}(\hat{\mu}, \hat{\nu}) = \max_{\omega\in\mathcal{H}:\|\omega\|_2\leq 1}\min_{\mathbf{P}\in\mathcal{P}_{\hat{\mu},\hat{\nu}}}\left[\sum_{i=1,j=1}^{m,n} P_{i,j}|\omega(x_i) - \omega(y_j)|^p\right]^{\frac{1}{p}} \qquad (45)$$

$$= \max_{\omega\in\mathcal{H}:\|\omega\|_2\leq 1}\left[\int_0^1 |F_{\omega,\hat{\mu}}^{-1}(q) - F_{\omega,\hat{\nu}}^{-1}(q)|^p dq\right]^{\frac{1}{p}}, \qquad (46)$$

where $\mathcal{P}_{\hat{\mu},\hat{\nu}} = \{\mathbf{P}\in[0,1]^{m\times n}|\mathbf{P}\mathbf{1}_n = \boldsymbol{\mu}, \mathbf{P}^\top\mathbf{1}_m = \boldsymbol{\nu}\}$ is a transportation polytope. The empirical distribution functions are $F_{\omega,\hat{\mu}}(w) = \sum_{i=1}^m\mu_i\mathbb{I}_{\omega(x_i)\leq w}$ and $F_{\omega,\hat{\nu}}(w) = \sum_{i=1}^n\nu_i\mathbb{I}_{\omega(y_i)\leq w}$ with inverses $F_{\omega,\hat{\mu}}^{-1}(q) = \min\{\omega(x_i), i \in \{1,\ldots,m\} : F_{\omega,\hat{\mu}}(\omega(x_i)) \geq q\}$ and $F_{\omega,\hat{\nu}}^{-1}(q) = \min\{\omega(y_i), i \in \{1,\ldots,n\} : F_{\omega,\hat{\nu}}(\omega(y_i)) \geq q\}$. For fixed $\omega$, the optimal transport plan $\hat{\mathbf{P}}$ is based on the sorted values $\omega(x_{(1)}) \leq \omega(x_{(2)}) \leq \cdots \leq \omega(x_{(m)})$ and $\omega(y_{(1)}) \leq \omega(y_{(2)}) \leq \cdots \leq \omega(y_{(m)})$. Denote the sorted values as vectors $\acute{\boldsymbol{\omega}}_X$ and $\acute{\boldsymbol{\omega}}_Y$, with $[\acute{\boldsymbol{\omega}}_X]_i = \omega(x_{(i)})$ and $[\acute{\boldsymbol{\omega}}_Y]_i = \omega(y_{(i)})$, and denote by $\acute{\boldsymbol{\mu}}$ and $\acute{\boldsymbol{\nu}}$ the corresponding permuted versions of $\boldsymbol{\mu}$ and $\boldsymbol{\nu}$.

In the case of equal samples sizes of uniform measure, $m = n$ and $\boldsymbol{\mu} = \frac{1}{m}\mathbf{1}_m$ and $\boldsymbol{\nu} = \frac{1}{n}\mathbf{1}_n$, elements in $\mathcal{P}_{\hat{\mu},\hat{\nu}}$ are scaled elements in the Birkhoff polytope (the set of doubly stochastic matrices), and the solution to the linear program is a permutation and $max\text{-}W_p^{\mathcal{H}}(\hat{\mu},\hat{\nu}) = \sup_{\omega \in \mathcal{H}:\|\omega\|_2 \leq 1} \left\{ \left[ \sum_{i=1}^m \left| \omega(x_{(i)}) - \omega(y_{(i)}) \right|^p \right]^{\frac{1}{p}} = \|\boldsymbol{\omega}'_X - \boldsymbol{\omega}'_Y\|_p \right\}$, where $\|\cdot\|_p$ denotes the $\ell_p$ norm in $m$ dimensions. As in the continuous case, the discrete transport plan between the sorted measures has the cumulative distribution $\acute{\mathbf{G}} \in [0,1]^{m \times n}$ with $\acute{G}_{i,j} = \min\{\sum_{k=1}^i \acute{\mu}_k, \sum_{k=1}^j \acute{\nu}_k\}$. The optimal transport plan between the sorted measures is given by taking the first difference over both rows and columns of $\acute{\mathbf{G}}$, $\acute{P}_{1,1} = \acute{G}_{1,1}$, $\acute{P}_{1,j} = \acute{G}_{1,j} - \acute{G}_{1,j-1}$, $\acute{P}_{i,1} = \acute{G}_{i,1} - \acute{G}_{i-1,1}$, and $\acute{P}_{i,j} = \acute{P}_{i,j} - \acute{P}_{i-1,j} - \acute{P}_{i,j-1}$. Using the sorted witness function evaluations the distance can be written as

$$max\text{-}W_p^{\mathcal{H}}(\hat{\mu},\hat{\nu}) = \max_{\omega \in \mathcal{H}:\|\omega\|_2 \leq 1} \left[ \sum_{i=1,j=1}^{m,n} \acute{P}_{i,j} \left| \omega(x_{(i)}) - \omega(y_{(j)}) \right|^p \right]^{\frac{1}{p}}. \qquad (47)$$

We now turn our attention to the optimization of the function $\omega$ parametrized in terms of $\boldsymbol{\alpha}$, $\omega(\cdot) = \sum_{i=1}^{m+n} \alpha_i \kappa(\cdot, z_i)$. For arbitrary sample sizes with $p = 2$, the max-sliced kernel W2 distance is

$$max\text{-}W_2^{\mathcal{H}}(\hat{\mu},\hat{\nu}) = \max_{\omega \in \mathcal{H}:\|\omega\|_2 \leq 1} \left( \|\omega\|_{\hat{\mu}}^2 + \|\omega\|_{\hat{\nu}}^2 - 2\max_{\mathbf{P} \in \mathcal{P}_{\hat{\mu},\hat{\nu}}} \langle \mathbf{P}, \boldsymbol{\omega}_X \boldsymbol{\omega}_Y^\top \rangle \right)^{\frac{1}{2}}, \qquad (48)$$

with unsorted values $\boldsymbol{\omega}_X = [\omega(x_1), \ldots, \omega(x_m)]^\top = \mathbf{K}_{XZ}\boldsymbol{\alpha}$ and $\boldsymbol{\omega}_Y = [\omega(y_1), \ldots, \omega(y_n)]^\top = \mathbf{K}_{YZ}\boldsymbol{\alpha}$.

The max-sliced kernel W2 distance can be expressed in terms of equation 45 or equation 48:

$$\left[ max\text{-}W_2^{\mathcal{H}}(\hat{\mu},\hat{\nu}) \right]^2 = \max_{\boldsymbol{\alpha}:\boldsymbol{\alpha}^\top \mathbf{K}\boldsymbol{\alpha} \leq 1} \min_{\mathbf{P} \in \mathcal{P}_{\hat{\mu},\hat{\nu}}} \sum_{i=1,j=1}^{m,n} P_{ij} \left| \sum_{k=1}^{m+n} (\kappa(x_i, z_k) - \kappa(y_j, z_k))\alpha_k \right|^2$$

$$= \max_{\boldsymbol{\alpha}:\boldsymbol{\alpha}^\top \mathbf{K}\boldsymbol{\alpha} \leq 1} \left\{ \boldsymbol{\alpha}^\top \mathbf{K}_{XZ}^\top \mathbf{D}_\mu \mathbf{K}_{XZ}\boldsymbol{\alpha} + \boldsymbol{\alpha}^\top \mathbf{K}_{YZ}^\top \mathbf{D}_\nu \mathbf{K}_{YZ}\boldsymbol{\alpha} \right.$$

$$\left. - 2\max_{\mathbf{P} \in \mathcal{P}_{\hat{\mu},\hat{\nu}}} \boldsymbol{\alpha}^\top \mathbf{K}_{XZ}^\top \mathbf{P}\mathbf{K}_{YZ}\boldsymbol{\alpha} \right\}$$

$$= \max_{\boldsymbol{\alpha}:\boldsymbol{\alpha}^\top \mathbf{K}\boldsymbol{\alpha} \leq 1} \min_{\mathbf{P} \in \mathcal{P}_{\hat{\mu},\hat{\nu}}} \boldsymbol{\alpha}^\top \mathbf{Q}_{\mathbf{P}}\boldsymbol{\alpha} = \max_{\boldsymbol{\alpha}} \frac{\min_{\mathbf{P} \in \mathcal{P}_{\hat{\mu},\hat{\nu}}} \boldsymbol{\alpha}^\top \mathbf{Q}_{\mathbf{P}}\boldsymbol{\alpha}}{\boldsymbol{\alpha}^\top \mathbf{K}\boldsymbol{\alpha}}, \qquad (49)$$

where $\mathbf{Q}_{\mathbf{P}} = \mathbf{K}_{XZ}^\top \mathbf{D}_\mu \mathbf{K}_{XZ} + \mathbf{K}_{YZ}^\top \mathbf{D}_\nu \mathbf{K}_{YZ} - \mathbf{K}_{XZ}^\top \mathbf{P}\mathbf{K}_{YZ} - \mathbf{K}_{YZ}^\top \mathbf{P}^\top \mathbf{K}_{XZ}$ is a symmetric matrix positive semidefinite matrix. (This can be seen since the objective is greater than or equal to zero for all choices of $\boldsymbol{\alpha}$.) For fixed $\mathbf{P}$, equation 49 is a convex maximization, that can be solved as a generalized eigenvalue problem. For fixed $\boldsymbol{\alpha}$, the optimization in terms of $\mathbf{P}$ is a linear program with the solution detailed above. However, when maximizing with respect to $\boldsymbol{\alpha}$, $\mathbf{P}$ is a matrix-valued function of $\boldsymbol{\alpha}$. To find a local maximum for $\boldsymbol{\alpha}$ we use the unconstrained optimization equation 49 and perform gradient ascent with respect to $\boldsymbol{\alpha}$, wherein each iteration we compute the optimal transport plan. This approach is also used in computing the generalized max-sliced Wasserstein distance Kolouri et al. (2019).

## A.5 COMPUTING THE MAX-SLICED DIVERGENCES FOR THE LINEAR CASE

The linear case follows from the kernel case with some further simplification. The objectives of the one-sided max-sliced Bures divergences are each a difference of convex functions, whose stationary points are maximum eigenvalue problems, and correspond to reweighted versions of the one-sided max-sliced TV divergence. Assuming the covariance matrices are strictly positive definite $max\text{-}D_B^{\mathbb{R}^d}(\hat{\mu},\hat{\nu}) = \max_{0 < \gamma < 1} \left| \sqrt{\mathbf{w}_\gamma^\top \boldsymbol{\rho}_X \mathbf{w}_\gamma} - \sqrt{\mathbf{w}_\gamma^\top \boldsymbol{\rho}_Y \mathbf{w}_\gamma} \right|$, where $\mathbf{w}_\gamma = \arg\max_{\mathbf{w}:\|\mathbf{w}\|_2 \leq 1} \mathbf{w}^\top (\gamma \boldsymbol{\rho}_X - \boldsymbol{\rho}_Y)\mathbf{w}$ for the one-sided case. If the matrices are singular, then cases where $\mathbf{w}$ is in the nullspace must be checked. Without loss of generality, the algorithm to obtain the optimal slice for the one-sided max-sliced Bures divergence is described in Algorithm 2.

---

**Algorithm 2:** One-sided max-sliced Bures divergence

**Input:** $\boldsymbol{\rho}_X, \boldsymbol{\rho}_Y \in \mathbb{R}^{d \times d}$

1  $\mathbf{w}_\gamma : \gamma \mapsto \arg\max_{\mathbf{w}:\|\mathbf{w}\|_2 \leq 1} \mathbf{w}^\top (\gamma \boldsymbol{\rho}_X - \boldsymbol{\rho}_Y) \mathbf{w}$

2  $\gamma^\star = \arg\max_{0 < \gamma \leq 1} \sqrt{\mathbf{w}_\gamma^\top \boldsymbol{\rho}_X \mathbf{w}_\gamma} - \sqrt{\mathbf{w}_\gamma^\top \boldsymbol{\rho}_Y \mathbf{w}_\gamma}$

3  **if** $\mathrm{rank}(\boldsymbol{\rho}_Y) = d$ **then**

4  $\quad$ $\mathbf{w}_{\mu > \nu} = \mathbf{w}_{\gamma^\star}$

5  **else**

6  $\quad$ $\mathbf{v} = \arg\max\limits_{\mathbf{w} \in \mathrm{Null}(\boldsymbol{\rho}_Y):\|\mathbf{w}\|_2 \leq 1} \mathbf{w}^\top \boldsymbol{\rho}_X \mathbf{w}$

7  $\quad$ **if** $\sqrt{\mathbf{v}^\top \boldsymbol{\rho}_X \mathbf{v}} > \sqrt{\mathbf{w}_{\gamma^\star}^\top \boldsymbol{\rho}_X \mathbf{w}_{\gamma^\star}} - \sqrt{\mathbf{w}_{\gamma^\star}^\top \boldsymbol{\rho}_Y \mathbf{w}_{\gamma^\star}}$ **then**

8  $\quad\quad$ $\mathbf{w}_{\mu > \nu} = \mathbf{v}$

9  $\quad$ **else**

10 $\quad\quad$ $\mathbf{w}_{\mu > \nu} = \mathbf{w}_{\gamma^\star}$

11 $\quad$ **end**

12 **end**

**Output:** $\mathbf{w}_{\mu > \nu}$

---

As an alternative to Algorithm 2, first-order algorithms can be applied, nevertheless, obtaining a global optimal cannot be guaranteed easily in this case. To make the objective differentiable, the square root, which is non-differentiable at 0, should be smoothed $\sqrt{\cdot} \approx \sqrt{\cdot + \epsilon^2}$ (e.g., $\epsilon^2 = 0.01$).

### A.6 SAMPLE-BASED MAX-SLICED BURES DISTANCE IS A RELAXATION OF THE MAX-SLICED WASSERSTEIN-2 DISTANCE

We show that the max-sliced Bures distance is a relaxation of the max-sliced W2 distance. Let $\boldsymbol{\rho} = \mathbf{Z}\mathbf{Z}^\top$ with $\mathbf{Z} \in \mathbb{R}^{d \times p}$ denote a strictly positive definite matrix, such that $\sqrt{\mathbf{w}^\top \boldsymbol{\rho} \mathbf{w}} > 0$, $\sqrt{\mathbf{w}^\top \boldsymbol{\rho} \mathbf{w}} = \sqrt{\mathbf{w}^\top \mathbf{Z}\mathbf{Z}^\top \mathbf{w}} = \max_{\theta \in \mathbb{S}^{p-1}} \langle \theta, \mathbf{Z}^\top \mathbf{w} \rangle$, where $\theta^* = \arg\max_{\theta \in \mathbb{S}^{p-1}} \langle \theta, \mathbf{Z}^\top \mathbf{w} \rangle = \frac{1}{\sqrt{\mathbf{w}^\top \mathbf{Z}\mathbf{Z}^\top \mathbf{w}}} \mathbf{Z}^\top \mathbf{w}$. Using this form, when $\boldsymbol{\rho}_X$ and $\boldsymbol{\rho}_Y$ are non-singular, the one-sided sliced Bures can be expressed as

$$\max_{\theta_1 \in \mathbb{S}^{d-1}} \langle \theta_1, \sqrt{\boldsymbol{\rho}_X} \mathbf{w} \rangle - \max_{\theta_2 \in \mathbb{S}^{d-1}} \langle \theta_2, \sqrt{\boldsymbol{\rho}_Y} \mathbf{w} \rangle = \max_{\theta_3 \in \mathbb{S}^{m-1}} \langle \theta_3, \mathbf{D}_{\boldsymbol{\mu}}^{\frac{1}{2}} \mathbf{X}^\top \mathbf{w} \rangle - \max_{\theta_4 \in \mathbb{S}^{n-1}} \langle \theta_4, \mathbf{D}_{\boldsymbol{\nu}}^{\frac{1}{2}} \mathbf{Y}^\top \mathbf{w} \rangle,$$

since $\boldsymbol{\rho}_X = \mathbf{X}\mathbf{D}_{\boldsymbol{\mu}}^{\frac{1}{2}}(\mathbf{X}\mathbf{D}_{\boldsymbol{\mu}}^{\frac{1}{2}})^\top$ and $\boldsymbol{\rho}_Y = \mathbf{Y}\mathbf{D}_{\boldsymbol{\nu}}^{\frac{1}{2}}(\mathbf{Y}\mathbf{D}_{\boldsymbol{\nu}}^{\frac{1}{2}})^\top$. Squaring this quantity and taking the square root yields the objective of the max-sliced Bures distance in a similar form to the max-sliced W2 distance,

$$max\text{-}D_B^{\mathbb{R}^d}(\hat{\mu}, \hat{\nu}) = \max_{\mathbf{w}:\|\mathbf{w}\|_2 \leq 1} \sqrt{\mathbf{w}^\top (\boldsymbol{\rho}_X + \boldsymbol{\rho}_Y) \mathbf{w} - 2 \max_{\mathbf{Q} \in \mathcal{Q}_{\hat{\mu}, \hat{\nu}}} \mathbf{w}^\top \mathbf{X}\mathbf{Q}\mathbf{Y}^\top \mathbf{w}}, \tag{50}$$

where $\mathcal{Q}_{\hat{\mu}, \hat{\nu}} = \{\mathbf{D}_{\boldsymbol{\mu}}^{\frac{1}{2}} \Theta \mathbf{D}_{\boldsymbol{\nu}}^{\frac{1}{2}} : \Theta \in \mathbb{R}^{m \times n}, \|\Theta\|_1 \leq 1\}$. This holds since

$$\max_{\Theta:\|\Theta\|_1 \leq 1} \mathbf{w}^\top \mathbf{X}\mathbf{D}_{\boldsymbol{\mu}}^{\frac{1}{2}} \Theta \mathbf{D}_{\boldsymbol{\nu}}^{\frac{1}{2}} \mathbf{Y}^\top \mathbf{w} = \max_{\theta_3 \in \mathbb{S}^{m-1}, \theta_4 \in \mathbb{S}^{n-1}} \langle \theta_3, \mathbf{D}_{\boldsymbol{\mu}}^{\frac{1}{2}} \mathbf{X}^\top \mathbf{w} \rangle \langle \theta_4, \mathbf{D}_{\boldsymbol{\nu}}^{\frac{1}{2}} \mathbf{Y}^\top \mathbf{w} \rangle \tag{51}$$

$$= \sqrt{\mathbf{w}^\top \boldsymbol{\rho}_X \mathbf{w}} \sqrt{\mathbf{w}^\top \boldsymbol{\rho}_Y \mathbf{w}}. \tag{52}$$

### A.7 COVARIATE SHIFT CORRECTION ALGORITHMS

For covariate shift correction the goal is to minimize the divergence by adjusting the weights $\boldsymbol{\nu}$ of the instances in one sample $\hat{\nu}$. This results in the following convex optimization problem:

$$\min_{\boldsymbol{\nu} \in \mathbb{R}_{\geq 0}^n : \langle \boldsymbol{\nu}, \mathbf{1} \rangle = 1} J(\boldsymbol{\nu}), \tag{53}$$

where

$$J(\boldsymbol{\nu}) = \left(max\text{-}D_B^{\mathcal{H}}(\hat{\mu}, \hat{\nu})\right)^2 = \max_{\boldsymbol{\alpha}:\boldsymbol{\alpha}^\top \mathbf{K}\boldsymbol{\alpha} \leq 1} \tilde{J}(\boldsymbol{\nu}, \boldsymbol{\alpha})$$

$$= \max_{\boldsymbol{\alpha}:\boldsymbol{\alpha}^\top \mathbf{K}\boldsymbol{\alpha} \leq 1} \underbrace{\langle \boldsymbol{\mu}, (\mathbf{K}_{XZ}\boldsymbol{\alpha})^{\circ 2}\rangle}_{C_{\boldsymbol{\alpha}}} + \langle \boldsymbol{\nu}, \underbrace{(\mathbf{K}_{YZ}\boldsymbol{\alpha})^{\circ 2}}_{\mathbf{k}_{\boldsymbol{\alpha}}}\rangle - 2\sqrt{\langle \boldsymbol{\mu}, (\mathbf{K}_{XZ}\boldsymbol{\alpha})^{\circ 2}\rangle \langle \boldsymbol{\nu}, (\mathbf{K}_{YZ}\boldsymbol{\alpha})^{\circ 2}\rangle}$$

$$= \max_{\boldsymbol{\alpha}:\boldsymbol{\alpha}^\top \mathbf{K}\boldsymbol{\alpha} \leq 1} C_{\boldsymbol{\alpha}} + \langle \boldsymbol{\nu}, \mathbf{k}_{\boldsymbol{\alpha}}\rangle - 2\sqrt{C_{\boldsymbol{\alpha}}}\sqrt{\langle \boldsymbol{\nu}, \mathbf{k}_{\boldsymbol{\alpha}}\rangle}.$$

For fixed $\boldsymbol{\alpha}$ the function $\tilde{J}(\cdot, \boldsymbol{\alpha})$ is a sum of a linear function and a convex function $f(\cdot) = -\sqrt{\cdot}$ is convex since $\sqrt{\cdot}$ is concave. $J(\boldsymbol{\nu}) = \max_{\boldsymbol{\alpha}:\boldsymbol{\alpha}^\top \mathbf{K}\boldsymbol{\alpha} \leq 1} \tilde{J}(\boldsymbol{\nu}, \boldsymbol{\alpha})$ is convex since maximizing over $\boldsymbol{\alpha}$ preserves convexity (Nesterov, 2018, Theorem 3.1.8).

In the linear kernel case, the cost is

$$J(\boldsymbol{\nu}) = \left(max\text{-}D_B^{\mathbb{R}^d}(\hat{\mu}, \hat{\nu})\right)^2 = \max_{\mathbf{w}:\|\mathbf{w}\|_2 \leq 1} \left\{ \tilde{J}(\nu, \mathbf{w}) = \left( \sqrt{\langle \boldsymbol{\mu}, (\mathbf{X}^\top \mathbf{w})^{\circ 2}\rangle} - \sqrt{\langle \boldsymbol{\nu}, (\mathbf{Y}^\top \mathbf{w})^{\circ 2}\rangle} \right)^2 \right\}$$

$$= \max_{\mathbf{w}:\|\mathbf{w}\|_2 \leq 1} (\langle \boldsymbol{\mu}, (\mathbf{X}^\top \mathbf{w})^{\circ 2}\rangle + \langle \boldsymbol{\nu}, (\mathbf{Y}^\top \mathbf{w})^{\circ 2}\rangle - 2\sqrt{\langle \boldsymbol{\mu}, (\mathbf{X}^\top \mathbf{w})^{\circ 2}\rangle \langle \boldsymbol{\nu}, (\mathbf{Y}^\top \mathbf{w})^{\circ 2}\rangle}.$$

Again, $J(\boldsymbol{\nu})$ is convex since $\tilde{J}(\cdot, \mathbf{w})$ is convex for fixed $\mathbf{w}$. In this case, the gradient has the intuitive form of

$$\nabla_{\boldsymbol{\nu}} J(\boldsymbol{\nu}) = \left(1 - \frac{\|\omega\|_{\hat{\mu}}}{\|\omega\|_{\hat{\nu}}}\right) (\boldsymbol{\omega}_{\mathbf{Y}}^\top)^{\circ 2} = \left(1 - \frac{\sqrt{\langle \boldsymbol{\mu}, (\mathbf{X}^\top \mathbf{w}_{\boldsymbol{\nu}})^{\circ 2}\rangle}}{\sqrt{\langle \boldsymbol{\nu}, (\mathbf{Y}^\top \mathbf{w}_{\boldsymbol{\nu}})^{\circ 2}\rangle}}\right) (\mathbf{Y}^\top \mathbf{w}_{\boldsymbol{\nu}})^{\circ 2}. \tag{54}$$

To solve this convex minimization over a probability simplex we apply the Frank-Wolfe (conditional gradient) algorithm Jaggi (2013) to iteratively adjust the weight of one instance $\boldsymbol{\nu} \leftarrow (1-\gamma)\boldsymbol{\nu} + \gamma \mathbf{e}_i$, where $\mathbf{e}_i$ is an indicator vector, $i = \arg\min_{1 \leq j \leq n}[\nabla_{\boldsymbol{\nu}} J(\boldsymbol{\nu})]_j$, and $\gamma \in [0, 1]$ is the stepsize. A benefit of the Frank-Wolfe scheme is that it requires only rank-1 updates of $\mathbf{Y}\mathbf{D}_{\boldsymbol{\nu}}\mathbf{Y}^\top$, which are needed for updating $\mathbf{w}$.

## A.8 ADDITIONAL EXPERIMENTAL RESULTS

We start by comparing the proposed max-sliced Bures distance to the max-sliced W2 distance for two-dimensional data. We compare the fixed-point algorithm for solving each one-sided max-sliced Bures divergence with gradient-based approaches for the max-sliced W2 distance using ADAM with parameters `lr = 1e-3, beta1 = 0.9, beta2 = 0.999, epsilon = 1e-08`, capping the number of iterations at 1000 or until the change in the slice is minimal $\|\mathbf{w} - \mathbf{w}_{\text{old}}\|_\infty < 10^{-6}$.

In two-dimensions, a near optimal slice can be obtained by a fine grid search of the sliced Bures and sliced W2 distance as shown in Figure 7 for two zero-mean Gaussian distributions. Figure 8 shows the cases for success rate of the gradient-based optimizations across 10 trials at varying sample sizes for 2- and 1000-dimensional zero-mean Gaussians. The effect of the number of gradient iterations is reported in Figure 9.

For kernel-based divergences, maximum mean discrepancy (MMD) detects differences in the first moments of the distributions in the RKHS. Using uncentered second-moments, the kernel-based max-sliced Bures distance (MSB) may detect some of the same differences. Figure 10 details witness function evaluations of each for six data sets generated from two-dimensional distributions, where a Gaussian kernel function is used. Notably, the one-sided MSB divergences correspond to localized regions, which are not distributed outliers. This is beneficial for the 'precision' of the witness function, but the 'recall' of MMD is better. This benefit of this localization depends on the task.

## A.9 COVARIATE SHIFT DETECTION WITH MAX-SLICED KERNEL DIVERGENCES

We proceed to generalize the comparisons in Figure 4 on imbalanced samples on MNIST to the kernel case. We use a Gaussian kernel $\kappa_\sigma$ with the parameter $\sigma$ set as the median Euclidean distance

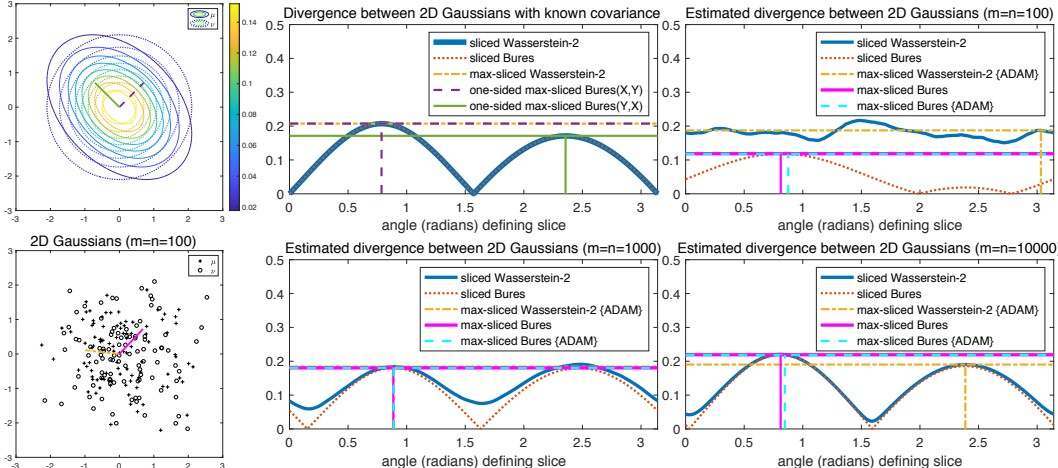

Figure 7: Sliced and max-sliced Bures and Wasserstein-2 distances are compared on population statistics and samples of varying sizes. $\mu = \mathcal{N}(\mathbf{0}, \mathbf{C})$ and $\nu = \mathcal{N}(\mathbf{0}, \mathbf{I})$, where $\mathbf{C} = \mathbf{Z}\mathbf{Z}^\top$, and $\mathbf{Z} \in \mathbb{R}^{2 \times 2}$ with entries that are originally standard normals and then row normalized such that $\mathbf{C}$ is a correlation matrix. In the population case and for zero-mean Gaussians, the Bures distance is equivalent to the W2 distance (Gelbrich, 1990). In the sample case, it is a lower bound. At both $m = 100$ and $m = 10^4$ the gradient optimization of the max-sliced W2 distance fails to obtain the global optimal slice (instead obtaining a local optimum).

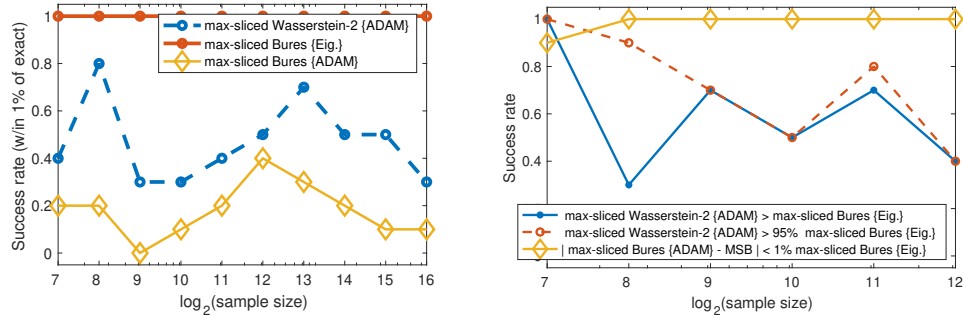

Figure 8: Success rate of finding optimal slices for the max-sliced Bures and W2 distances across samples of varying sizes (10 random runs per size). The distributions are zero-mean Gaussians, with $\mu = \mathcal{N}(\mathbf{0}, \mathbf{C})$ and $\nu = \mathcal{N}(\mathbf{0}, \mathbf{I})$, where $\mathbf{C} = \mathbf{Z}\mathbf{Z}^\top$, and $\mathbf{Z} \in \mathbb{R}^{d \times d}$ with entries that are originally standard normals and then row normalized such that $\mathbf{C}$ is a correlation matrix. (Left) In the case of $d = 2$, success is obtained for a distance within 1% of the value obtained by fine-grid search of angles. In this case, the gradient approach for the max-sliced Bures fails more often than the max-sliced W2. (Right) For $d = 1000$ the eigenvalue-based approach (Algorithm A.5) defines the global optimum. In the larger dimension, the gradient approach for the max-sliced Bures {ADAM} succeeds in almost all of the cases, within 1% of the optimal value obtained by Algorithm A.5 (MSB), whereas the max-sliced W2 distance fails to upper bound the max-sliced Bures on roughly half the trials.

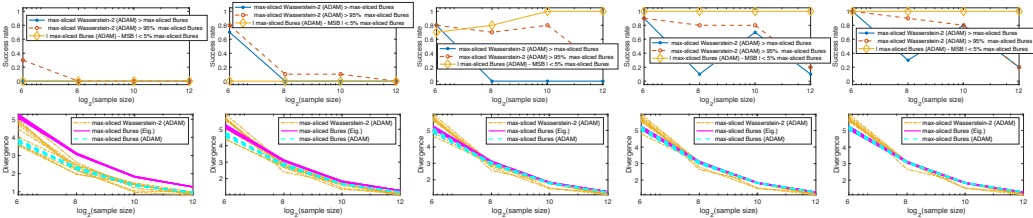

Figure 9: Performance of gradient algorithms for max-sliced Bures and max-sliced W2 distances across $d = 1000$ dimensional samples of varying sizes (10 random runs per size) and number of iterations in ADAM (Left to right: 50, 100, 200, 500, 1000). The samples are from zero-mean Gaussians distributions, with $\mu = \mathcal{N}(\mathbf{0}, \mathbf{C})$ and $\nu = \mathcal{N}(\mathbf{0}, \mathbf{I})$, where $\mathbf{C} = \mathbf{Z}\mathbf{Z}^\top$, and $\mathbf{Z} \in \mathbb{R}^{d \times d}$ with entries that are originally standard normals and then row normalized such that $\mathbf{C}$ is a correlation matrix. (Top) For the max-sliced W2 a successful run is obtained when it is greater than or equal to the optimal solution to the max-sliced Bures (blue solid line) or when it is greater than 95% of max-sliced Bures (red dotted with circles). For the gradient approach to max-sliced Bures, success is when the difference to the optimal is 5% of the optimal (yellow solid with diamonds). (Bottom) Divergence values obtained across the 10 trials with increasing sample size.

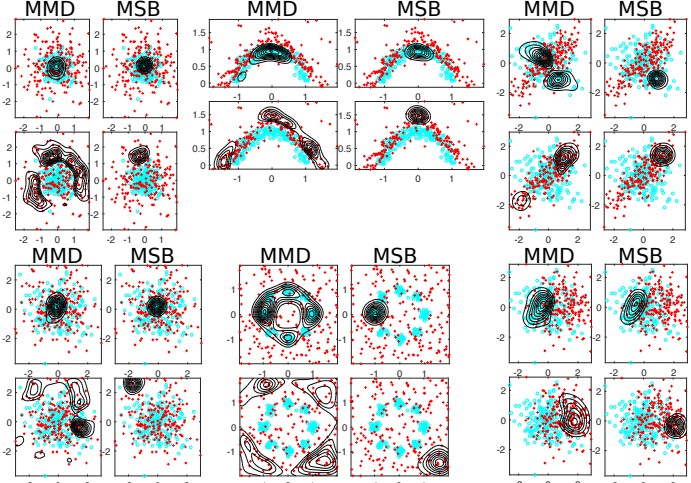

Figure 10: Maximum mean discrepancy (MMD) and max-sliced Bures distance (MSB) applied to two-dimensional samples using a Gaussian kernel. For each data set (shown as a two-by-two subplot), the contour plots indicate the squared magnitude of the witness function evaluations. For MMD, positive witness function values are plotted in the top row and negative evaluations are in the second row. For MSB, the rows correspond to the two one-sided divergences. The witness functions for the one-sided MSB divergences correspond to localized regions.

in the pooled sample. To ease computation for large-sample sizes, we let $\tau$ be a random subset of the pooled samples $\{x_i\}_{i=1}^m \cup \{y_i\}_{i=1}^n$ of size $l = \min\{500, m+n\}$. In this case, the max-sliced Fréchet refers to $max_L\text{-}D_{GW}^{\mathcal{H}}$, which is equal to the square root of the sum of square of MMD and the square of the max-sliced Bures using centered kernels, as in equation 23. The kernel-based max-sliced W2 distance should be an upper bound of the max-sliced Fréchet. However, in practice the optimal slice (witness function) may not be obtained.

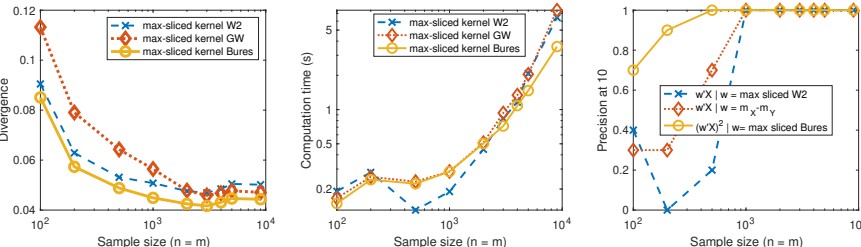

Figure 11: Kernel-based max-sliced distances are applied to balanced and imbalanced samples from MNIST. The first sample $\hat{\mu}$ consists of the training set (balanced classes with size $m$), and the second sample $\hat{\nu}$ is a $n$-sized sample from the test set with a minority class $l \in \{0, \ldots, 7\}$ with prevalance of 5%. (Left) Divergence estimates for increasing sample size for $l = 7$. Notably, for $m < 2000$ the max-sliced Wasserstein-2 distance fails to obtain the optimal slice as it should upper bound the max-sliced kernel Gauss-Wasserstein (Fréchet) distance. (Center) Corresponding computation time. (Right) Each curve is the average precision@10 (averaged across the 10 classes). The witness function for the one-sided max-sliced Bures $\omega_{\hat{\mu}>\hat{\nu}}$ can be used to reliably identify instances from $\hat{\mu}$ associated to the missing class.

We now compare the proposed kernel-based max-sliced divergences to existing baselines. A primary baseline for this task is to train a logistic regression model with kernel basis functions to distinguish the two samples, and then use the probability estimates of the instances as the witness function evaluations $\omega(x) = \Pr(H_0|X = x) = 1 - \Pr(H_1|X = x)$, where $H_0 : X \sim \hat{\nu}$ and $H_1 : X \sim \hat{\mu}$. As additional baselines we also tested three methods for importance reweighting and density ratio estimation: kernel mean matching (KMM) (Huang et al., 2007), least-squares importance estimation (uLSIF) (Kanamori et al., 2009), and relative density-ratio estimation (RuLSIF) (Yamada et al., 2011), but all methods were outperformed by logistic regression with kernel bases. We also compare with kernel Fischer discriminant analysis (KFDA) (Mika et al., 1999), and the linear cases of max-sliced Wasserstein-2 distance, its first moment approximation, max-sliced Bures, and logistic regression. For all kernel methods, a Gaussian kernel $\kappa_\sigma$ is used with the parameter $\sigma$ set as the median Euclidean distance in the pooled sample.

Using the MNIST data set again, we test three scenarios of covariate shift. For each, one sample has a mismatched probability for one class $l \in \{0, \ldots, 9\}$ and the other sample has a balanced sample: (Scenario 1) $\hat{\mu}$ is balanced and $\hat{\nu}$ is missing $l$; (Scenario 2) $\hat{\mu}$ is imbalanced with $l$ only appearing in 2% of the cases, compared to $10.\bar{8}\%$ for the other classes and $\hat{\nu}$ is balanced; (Scenario 3) $\hat{\mu}$ is balanced and $\hat{\nu}$ consists of only images from $l$. In each case, $\hat{\mu}$ is a sample of 500 images from the training set and $\hat{\nu}$ is a sample of 500 images from the test set. A threshold-free way to assess covariate shift detection is to use the area-under-the-curve (AUC) of the receiver operator curve (ROC), where positive instances correspond to class $l$. For some methods, the witness function (or its magnitude) may be ambiguous in sign, i.e., the values may be high (or large) for either the under- or over-sampled instances (namely, max-sliced W2). To be generous, on each run we choose the ordering with the highest AUC. The results are reported in Table 3.

The other baselines KMM, uLSIF, and RuLSIF are not shown (their AUC scores across the scenarios are worse than the logistic regression with kernel baseline). In a separate set of runs we also compute the realism scores (Kynkäänniemi et al., 2019) with $k = 3$ where $\hat{\nu}$ is considered the real set, and $\hat{\mu}$ are synthetic, to prioritize instances; results for the three scenarios are $0.75\pm0.11$, $0.67\pm0.12$, and $0.94\pm0.04$, which is better than linear logistic regression and KFDA, but far worse than the kernel logistic regression baseline.

Table 3: Unsupervised covariate shift outlier detection on MNIST. The goal is to identify instances associated with an over- or underrepresented class $l \in \{0, \ldots, 9\}$. Values are AUC where positives are instances from class $l$. We report the mean and standard deviation and the number of times each method has the highest AUC (including ties) across 100 trials (10 for each case $l \in \{0, \ldots 9\}$).

|  | (Scenario 1) | (Scenario 2) | (Scenario 3) | (1) | (2) | (3) |
|---|---|---|---|---|---|---|
| logistic regression-linear | 0.60 (0.05) | 0.60 (0.08) | 0.91 (0.06) | 0 | 1 | 0 |
| Max-Sliced Bures | 0.89 (0.12) | 0.86 (0.13) | 0.95 (0.03) | 2 | 3 | 3 |
| Max-Sliced W2 (approx.) | 0.86 (0.10) | 0.84 (0.13) | 0.96 (0.02) | 1 | 2 | 0 |
| Max-Sliced W2 | 0.87 (0.14) | 0.85 (0.14) | 0.96 (0.02) | 2 | 12 | 0 |
| logistic regression-kernel | 0.90 (0.07) | 0.86 (0.10) | 0.99 (0.01) | 10 | 15 | 93 |
| KFDA | 0.58 (0.03) | 0.61 (0.12) | 0.91 (0.03) | 0 | 2 | 1 |
| MMD | 0.87 (0.10) | 0.85 (0.12) | 0.97 (0.02) | 1 | 3 | 0 |
| Max-Sliced Kernel TV | 0.85 (0.10) | 0.83 (0.14) | 0.96 (0.03) | 2 | 0 | 1 |
| Max-Sliced Kernel Bures | 0.92 (0.11) | 0.88 (0.14) | 0.97 (0.02) | 21 | 32 | 2 |
| Max-Sliced Kernel W2 | 0.92 (0.11) | 0.88 (0.13) | 0.97 (0.02) | 61 | 35 | 0 |

## A.10 COVARIATE SHIFT CORRECTION FOR CLASS-CONDITIONAL SUBSAMPLING

Figure 12 shows 20 synthetic images—generated by AutoGAN trained on CIFAR10 ($n$=50,000)—with the largest weights after reweighting in order to minimize the max-sliced Bures distance to the subset of training images for each class separately ($m$=5,000). Computing the max-sliced Bures distance with the entire training set of 50,000 points is tractable since it does not depend on the sample size. The realism scores of the selected images have a median and range of 0.96 (0.63–1.34). Figure 13 shows the same but based on the weights optimized by using the W2 distance with the mini-batch optimization as the cost function. The realism scores of the selected images have a median and range of 1.1 (0.93–1.29). Figure 14 shows the same but based on the weights optimized by using the max-sliced W2 distance with the mini-batch optimization. The realism scores of the selected images have a median and range of 0.97 (0.65–1.25). Finally, Figure 15 shows the synthetic images selected for having the highest realism scores; notably this set lacks class correspondence.

The optimizations in the first three cases use the Frank-Wolfe algorithm (Jaggi, 2013) with simplex constraints. The default step-size schedule $\gamma = \frac{2}{k+2}$ and the same stopping criterion is used $\max_{1 \leq i \leq n} |\nu_i^{(k)} - \nu_i^{(k-1)}| < 10^{-3}$, where $k$ is the iteration index. This yields roughly the same number of iterations for each method. The optimization starts from a uniform weighting, which means the weights for only ~2000 instances are actually individually adjusted (the rest are adjusted by common scaling). The Fréchet Inception distances after reweighting are detailed in Table 4. Based on the quantitative and qualitative results it appears that the W2 distance with mini-batch approximation assigns high weight to high-quality synthetic images, but the diversity of the highly weighted instances may not capture the full distribution for a class. In this regard, the max-sliced Bures better captures the diversity of the class, albeit choosing less realistic images.

Table 4: Fréchet Inception distances (FID) between CIFAR10 test set images in each class and reweighted sample of synthetic images from AutoGAN. The second column shows the FID to the corresponding training set. The third column is a uniform weighting over all 50,000 synthetic images. The reweighting that minimizes the max-sliced Bures distance (MSB) to the subset of training images performs the best on average. Using the W2 distance—estimated through 10 mini-batches of 100 images on each iteration—performs best only on one-class. The max-sliced W2 (MSW2) distance also uses mini-batches. The realism scores $R$ of the 20 images with the highest weight for each class (200 images for each method) are summarized by the median and range.

|  | Training set | Uniform | MSB | W2 | MSW2 |
|---|---|---|---|---|---|
| airplane | 28.00 | 108.67 | **57.82** | 74.67 | 81.85 |
| automobile | 20.20 | 133.10 | **39.88** | 66.76 | 74.81 |
| bird | 30.25 | 86.23 | 58.71 | **57.17** | 76.98 |
| cat | 35.53 | 84.63 | **61.27** | 83.89 | 75.57 |
| deer | 26.46 | 85.55 | **46.81** | 60.21 | 56.44 |
| dog | 28.99 | 106.76 | **56.88** | 67.94 | 78.77 |
| frog | 29.88 | 107.88 | **51.14** | 75.23 | 66.93 |
| horse | 24.33 | 111.30 | **42.68** | 66.77 | 63.24 |
| ship | 21.49 | 131.92 | **37.35** | 65.05 | 74.47 |
| truck | 17.77 | 141.56 | **38.97** | 63.73 | 76.05 |
| Average | 26.29 | 109.76 | **49.15** | 68.14 | 72.51 |
| Median $R$ |  |  | 0.96 | 1.10 | 0.97 |
| Range $R$ |  |  | 0.63–1.34 | 0.93–1.29 | 0.65–1.25 |

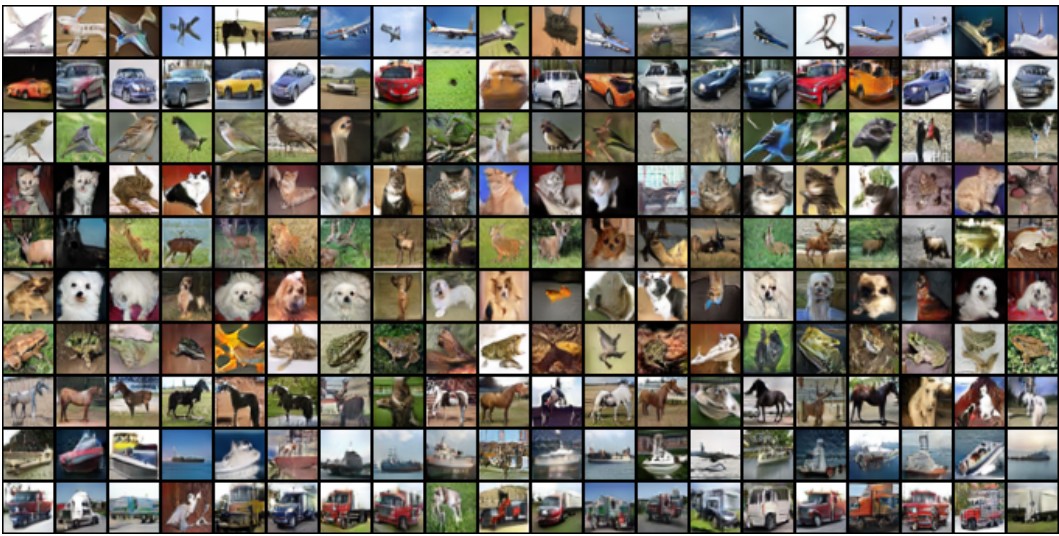

Figure 12: Distribution matching based on minimizing max-sliced Bures distance $max\text{-}D_B^{\mathbb{R}^d}(\hat{\mu}, \hat{\nu})$. Synthetic images shown are those with the highest values of $\nu$, where $\hat{\nu}$ is the $\nu$-weighted distribution over 50,000 synthetic images from AutoGAN and $\hat{\mu}$ consists of CIFAR10 training images for a single class in each row. Rows (top to bottom) correspond to airplane, automobile, bird, cat, deer, dog, frog, horse, ship, and truck classes.

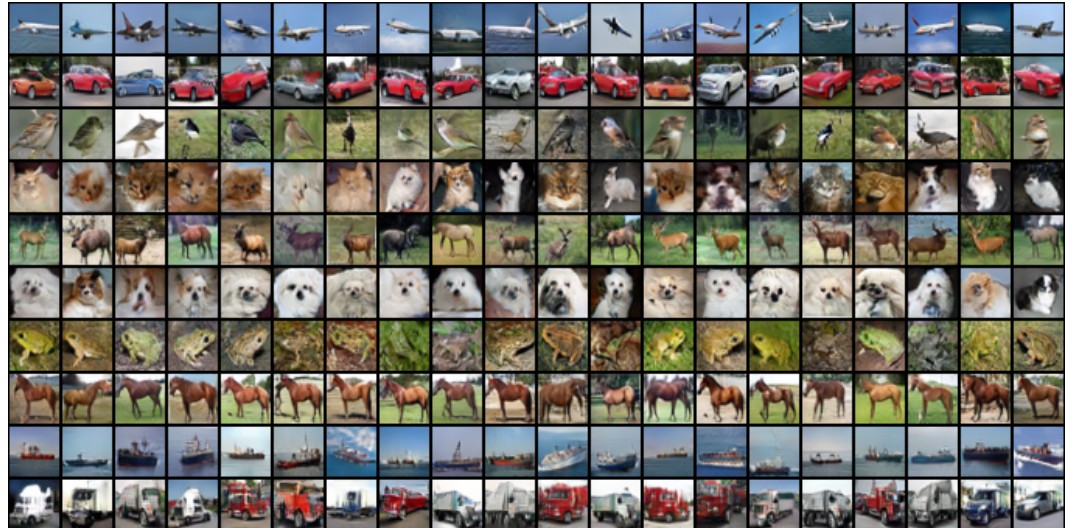

Figure 13: Distribution matching based on minimizing the Wasserstein-2 distance through mini-batch. Synthetic images shown are those with the higest values of $\boldsymbol{\nu}$, where $\hat{\nu}$ is the $\boldsymbol{\nu}$-weighted distribution over 50,000 synthetic images from AutoGAN and $\hat{\mu}$ consists of CIFAR10 training images for a single class in each row. Rows (top to bottom) correspond to airplane, automobile, bird, cat, deer, dog, frog, horse, ship, and truck classes.

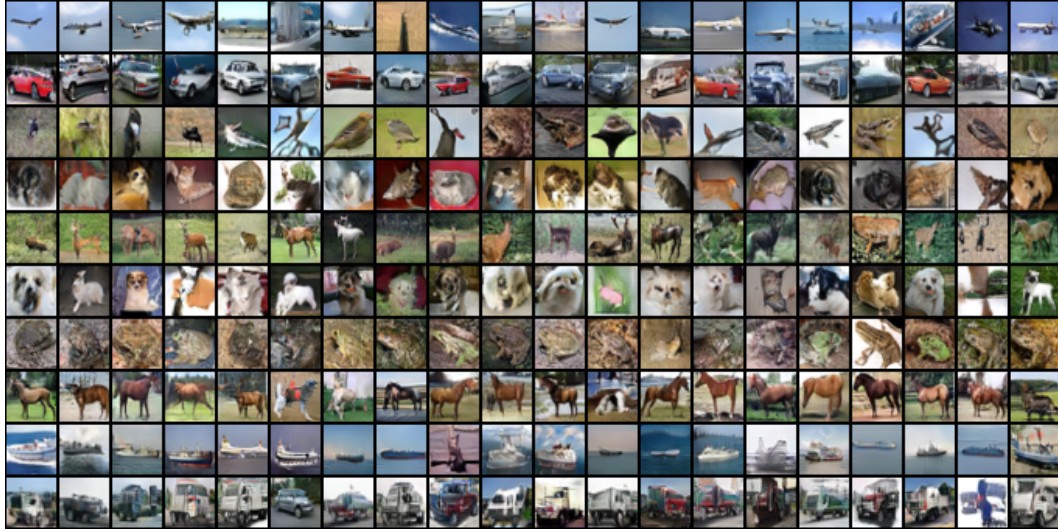

Figure 14: Distribution matching based on minimizing the max-sliced Wasserstein-2 distance $max\text{-}W_2^{\mathbb{R}^d}(\hat{\mu}, \hat{\nu})$ through mini-batch approximation. Synthetic images shown are those with the highest values of $\boldsymbol{\nu}$, where $\hat{\nu}$ is the $\boldsymbol{\nu}$-weighted distribution over 50,000 synthetic images from AutoGAN and $\hat{\mu}$ consists of CIFAR10 training images for a single class in each row. Rows (top to bottom) correspond to airplane, automobile, bird, cat, deer, dog, frog, horse, ship, and truck classes.

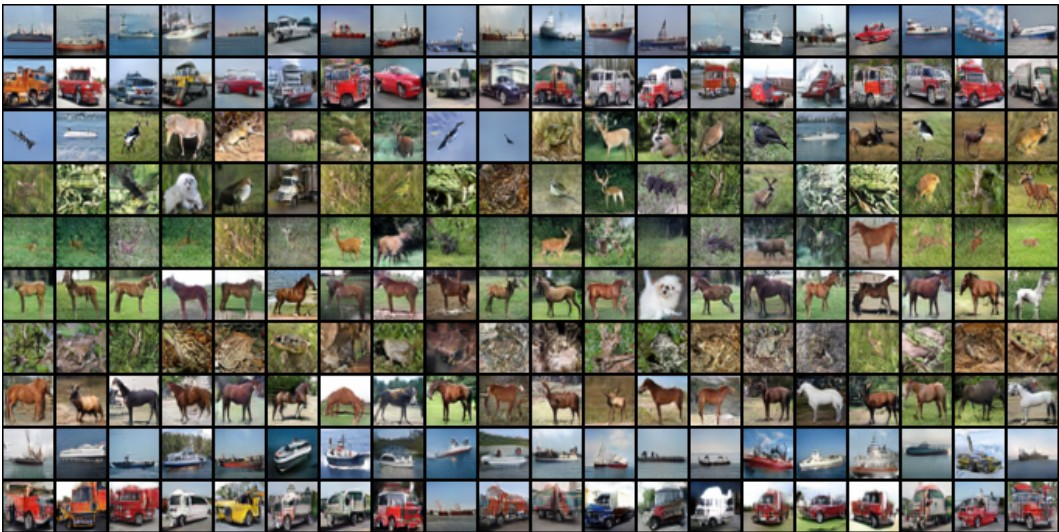

Figure 15: Selecting images directly with the highest realism scores. Synthetic images shown are those with the highest realism values over 50,000 synthetic images generated by AutoGAN when the realism scores used in each row are computed using the CIFAR10 training images for a single class. Rows (top to bottom) correspond to airplane, automobile, bird, cat, deer, dog, frog, horse, ship, and truck classes. Realism score correctly identifies "realistic" imagery, but is unable to find samples that cover the real distribution. For example, the top row is missing airplanes, the third row is missing birds, the fourth row is missing cats, etc.

