# OpenReview forum: "Max-sliced Bures Distance for Interpreting Discrepancies"
_ICLR.cc/2021/Conference — Reject_

### Official Review · AnonReviewer4 · 2020-10-29
**Simple extension of sliced Wasserstein distance, with some interesting ideas but lacking in motivation, clarity and evaluation**

**Rating:** 5
**Confidence:** 3

**Review:**

Summary:

This paper proposes a sliced version of the Bures distance, which is a lower bound on the 2-Wasserstein distance. The purpose behind this is to identify instances that are have the highest contribution towards the discrepancy between two distributions. But compared to other sliced OT distances, this one operates on a the Bures lower bound, which yields a more tractable solution. The paper presents experimental results on image classification datasets, which are claimed to show an advantage of the proposed variant over the full sliced Wasserstein distance .

Strengths:
* Timely and relevant problem: designing faster, sliced versions of OT distances is a very active area of research
* Some interesting ideas, and lots of connections to other distances

Weaknesses:
* Limited novelty. This paper heavily builds on recent work on sliced wasserstein distances (e.g. Kolouri), and combines it with recent work on OT on RKHS (Zhang et al)
* Clarity/Intuition. Various aspects of the paper could be better motivated, discussed and analyzed (see below).
* Unconvincing / confusing experimental evaluation, misses necessary quantitative experiments and includes two many qualitative ones

Detailed comments:
* The original sliced and max-sliced Wasserstein distances are never introduced here, which is surprising given how closely related they are. I would suggest prioritizing this over other (perhaps unnecessary) aspects of the background present in sections 2.1 and 2.2.
* The introduction of the proposed distances (6, 7, 8) could be better motivated. E.g., it is not immediately clear why (7) corresponds to a sliced version of (4). Going through the derivations in the Appendix sheds some light on this, but given that the main contribution of the paper is all contained in section 2.3, one would expect a more thorough discussion and presentation of this contribution
* Introducing RKHS and then falling back to finite dimensional embeddings (2.5.1) feels like an overkill. Also, under this setting, how is this different from just embedding all samples as a pre-processing step and then using the usual sliced Wasserstein?
* As for any lower/upper bound on a distance, the first question that the experiments should try to answer is: "how tight / good approximation is this bound"? Such an experiment is necessary to gauge the approximation-quality tradeoff that such a lower bound entails. In this case, I would have expected to see a vis-a-vis comparison of the three proposed distances, and the original sliced Wasserstein ones, on synthetic datasets of increasing size, to get a sense of their asymptotic behavior
* It might be just me, but I find the setup of the experiments in 3.1 very confusing, and have a hard time parsing the results. E.g. why is the one-sided version of the m-s-Bures distance used here? Is the plot on the right the usual sliced Wasserstein distance or the kernel one? Why is a first-moment approximation used? What is the relevance of the x-axis used here?
* A good chunk of the experimental results are purely qualitative and rely on the user comparing specific instances of images (Fig 3 and 4). Leaving aside the fact that the size/resolution is pretty small, I'm not sure what I'm looking for here. What should one expect to see here? Are these results good or bad?
* The covariate shift detection experiments suffer from a similar lack of explanation for the various experimental design choices (e.g., why/how were these scenarios chosen?). In addition, the results exhibit very very wide s.d. confidence intervals, to the point that all the max-sliced distances have intersecting intervals, making it almost impossible to draw any statistically significant results from this table.

---

> ### Author Response · Authors · 2020-11-23
> **Kernels improve the performance of sliced distances, max slicing can be interpreted as witness functions, and heuristic often fail to achieve bounds.**
>
> Thank you for your feedback, which has motivated a number of key revisions.
>
> We now state the expression for the max-sliced Wasserstein-p distance, and note the relations between linear and kernel versions.  We have reorganized other aspects of the background to clarify the contributions, which stem from the difficulty in obtaining an optimal slice for the max-sliced Wasserstein-p distance. A deficit addressed by a completely distinct approach in another submission: https://openreview.net/forum?id=QYjO70ACDK . As for the sliced Wasserstein,  we feel it does not fit the narrative as it does not provide an interpretable witness function.
>
> We have further clarified the derivations in the Appendix, but a lack of space prevents their inclusion in the main body. Nonetheless, we have clarified some of the notation regarding the root mean square of the witness function evaluations (RMS) $\lVert \omega \rVert_\mu =\sqrt{\langle \omega,\rho_X \omega\rangle_\mathcal{H}} = \lVert \sqrt{\rho_X }\omega\rVert_2$, which should be insightful to the reader.
>
> With regard to using kernels, we now introduce the linear case earlier for clarity. The reviewer is correct that embeddings such as random Fourier basis can be used to approximate the kernel approach (as shown in Section 3.3 and Figure 5). However, the elegance of the IPM-like approach is best seen through the functional perspective. Furthermore, the kernel-based measures prove more powerful for the covariate shift detection on MNIST.
>
> With slicing, the divergence is calculated on a one-dimensional subspace.  Max-sliced Wasserstein (without a non-linear kernel) is not able to optimize a reweighting in the one-dimensional subspace because after projection points near the origin never contribute to the divergence, yet they may not be well matched. This is shown in Figure 5. Using a kernel enables the resulting slice to be a non-linear function.
>
> With regard to lower/upper bounds, this is an important point, and we have now done new experiments in this direction. It turns out that the max-sliced Bures distance enables computation of the max-sliced Frechet distance that is even better than the estimator based on the max-sliced Wasserstein-2 distance. Specifically, our estimator for the lower bound is often above and closer to the true max-sliced Wasserstein-2 distance. We check this in the case of zero-mean multivariate normal distributions where the Bures distance coincides with the Wasserstein-2 distance. (The MATLAB scripts are released in the supplementary material).
>
> As for quantification of performance we have included performance metrics for the reweighting that were previously only in the Appendix. This shows that the proposed max-sliced Bures based reweighting minimizes the Fréchet Inception distance for each class of CIFAR10. We also added experiments examining a GAN trained on stacked MNIST to quantify the ability of the proposed distance to detect mode dropping throughout training. The code for this is released. (We also note that the basic GAN criticism experiment captures details regarding the values of the Realism scores which quantify the fake image quality.)
>
> The one-sided divergences give the user control of whether the witness function is expected to have high magnitude values in one distribution versus the other. This is explained in equations 1 and 2 and Figure 1. Thus, unlike the max-sliced Wasserstein the user will know ahead of time in what regions the witness function takes large values.  The first-moment surrogate in the max-sliced Wasserstein-2 distance is used since (i) it has a closed form solution, and (ii)  that is what was originally proposed by the authors Deshpande et al, (2019). We have now included more results comparing the max-sliced Bures to the max-sliced Wasserstein-2 using the approach from Kolouri et al. (2019).
>
> We would like to further mention that the two slices can each be applied to instances from either distribution.  In covariate shift detection, one is often interested in what test set instances $Y\sim \nu$ are underrepresented in training $X\sim \mu$. In this case we assume we only know the labels of instances in the training set $\hat\mu$. The user could then note which classes are underrepresented in training and try to compensate. We quantify performance of the witness function $\omega_{\mu>\nu}$ by checking if the instances with the largest magnitude witness function evaluations (witness points) in the training set are indeed from a minority class. This is shown in Figure 4.
>
> In criticizing a GAN one is interested whether the synthetic instances produced $Y\sim \nu$ are representative of the training set $X\sim\mu$.  In Section 3.1 (Figure 2), we quantify the performance of detecting different degrees of underrepresentation using $\omega_{\mu>\nu}$ applied to instances from $\nu$.  As shown in Figure 3, there are four sets of witness points: real and fake. The latter's quality is quantified by their realism scores.

---

### Official Review · AnonReviewer2 · 2020-10-29
**Good paper but the exposition is difficult to follow**

**Rating:** 6
**Confidence:** 2

**Review:**

This paper studies a family of integral probability metric (IPM) divergence on Hilbert spaces. This family can be characterized by the choice of the witness function, and specific witness function may give rise to the Bures distance, the MMD, Wasserstein, as well as many sliced variants. While this family has been well understood for distributions on finite dimensional space, this paper extends this insight to distributions on (possibly infinite dimensional) Hilbert space. By leveraging the representer theorem, the paper provides the finite dimensional optimization problems that can be solved to estimate the divergence from samples. The power of the method is demonstrated on the covariate shift experiments.

Positive points:
1. Extending previous results of IPM-type distance to the Hilbert space is a natural ideas. There are in fact many applications that can be nicely solved in the RKHS framework, and thus novel divergences that can substitute the MMD is desirable. The results can boost further learning tasks, and thus are relevant to the machine learning community.

Negative points:
I find that the exposition of this paper is extremely difficult to follow, and this downplays the contributions of this paper.

1. The title is misleading: the title implies that the paper will study the max-sliced Bures distance in-depth. However, the content of the paper is a general (more width and depth) on a general family of IPM on RKHS.
2. All results of this paper is provided in Section 2, and there is a tremendous difficulty to identify which results are from the literature, and which results are new in this paper.
3. From the theoretical viewpoint, it is still not clear to the reader why a divergence on a Hilbert space is necessary. Is it because of the possibility to have better sample complexity?

I think that the paper contains good results, but the paper needs to be thoroughly restructured. Currently, the main results of the papers are included in the Appendix. These results (theorems) should be streamlined into the main text. The introduction should be rewritten with a clear exposition of the contributions.

Minor comments:
- In page 5, the authors claim that a "local ascent algorithm will often yield the global optimum". Can the authors clarify how "often" is this? Otherwise, I would recommend to refrain from making unquantifiable claims.
- The first two lines in page 6 is difficult to parse.
- The word "optimizations" should be replaced by "optimization problems"

---

> ### Author Response · Authors · 2020-11-23
> **Revamped exposition with explanatory figures with examples; a tractable algorithm to obtain optimal slice**
>
> Thank you for your comments. They were extremely helpful for improving the paper.
>
> Firstly, we have reorganized the exposition. The introduction now balances the background with the contribution, and provides more insights (with main results moved from the Appendix). We believe the revised version provides a clear explanation of the methodology and the experiments. We still focus on the max-sliced Bures, but include the max-sliced total variation and max-sliced Wasserstein-2 as they provide necessary context. In particular, a weighted version of the max-sliced total variation is necessary to understand the optimality of the new algorithm (describes subsequently) that obtains the optimal slice for the one-sided max-sliced Bures divergence.
>
> Secondly, after some investigation we now have a tractable algorithm to compute the one-sided max-sliced Bures divergence that is guaranteed to find a global optima of the objective. The details of the solution are given in the Appendix along with extensive experiments for zero-mean Gaussians where the population versions of the Bures and Wasserstein-2 distance align. This allows us to quantify how frequently local ascent approaches for both max-sliced Bures and max-sliced Wasserstein-2 find the optimal slice. In 1000-dimensional space, iterative algorithms for the max-sliced Wasserstein-2 distance often yield distance values that are lower than the max-sliced Bures. This is suboptimal since max-sliced Bures is a lower bound on the max-sliced Wasserstein-2 distance. The rate of success depends on the number of iterations.

---

### Official Review · AnonReviewer5 · 2020-11-09
**Interesting work on relevant problem, but I am left desiring more compelling applications**

**Rating:** 7
**Confidence:** 3

**Review:**

##########################################################################

Summary: This work proposes the max-sliced Bures (MSB) distance, a distance metric for comparing probability distributions. This work adds to the existing literature on transport based slicing techniques for comparing probability distributions, such as sliced-Wasserstein (SW), max-sliced-Wasserstein (MSW), generalized sliced Wasserstein (GSW).

Novel applications in (1) interpreting datasets and (2) critiquing generative models are claimed due to the assignment of energy-based scores to instances, which allows identification of specific subsets that are not well-matched.

The authors present several experiments demonstrating their technique
(1) detecting class-imbalance [Figure 2]
(2) identifying under/over-represented data subsets in GAN generative models [Figure 3]
(3) detecting "covariate shift outlier detection" [Table 2]
(4) identifying covariate-shift through reweighting on synthetic CIFAR-10 [Figure 4]
(5) distribution matching on toy distributions (grid of 25 Gaussians) [Figure 5]

##########################################################################

Reasons for score:

Overall I vote for acceptance, but with some reservations on the experiments.

I found the mathematical analysis to be interesting and a potentially valuable contribution to the ML literature on probability distance. I did not check the math in detail, but the authors appears to know what they are talking about.

I was intrigued by the claimed applications. The technique is sold as a tool for (1) interpreting datasets (2) critiquing generative models. The community could benefit from more such tools.

At first read I found the empirical results lacking, but after a second reading I concluded they are reasonable. As I understand it, one difficulty that arises in interpreting the experiments is their novelty, i.e. at present there is no standard set of experiments for these applications. This seems unavoidable, but I have still have some doubts whether these experiments are optimal.

What I was hoping to see was a compelling demonstration of the technique on common problem arising in the practice of machine learning.  For instance in the conclusion the authors write "Additionally, the one-sided max-sliced Bures divergences ... can be used to identify systematic discrepancies such as **mode dropping**" [emphasis mine].


##########################################################################

Pros:
* Interesting mathematics
* Important problem

##########################################################################

Cons:
* Some difficulty in interpreting experiments
* More compelling applications (e.g. solving mode-dropping are desired)
* Comparison with other methods where possible seems weak (Figure 5)

##########################################################################

Questions during rebuttal period:

Mode-dropping is a very common problem. Can the authors show an example on which their technique identifies and helps solve mode dropping?

Figure 5 compares max-sliced Bures (cases C,D) against max-sliced W2 (case A,B) on matching a grid of Gaussians starting from uniform, an example taken from Kolouri et al. (2019). Qualitatively, case D beats case B. Why didn't the authors consider the other cases (8 circular Gaussians, swiss-roll, half-moons, circle) as in Figure 2 Kolouri et al. (2019)?

#########################################################################

Additional Feedback:

A code release will  improve this submission.

#########################################################################

POST-REBUTTAL RESPONSE:

I found the author's further work on (1) the mode-collapse experiments (2) quantitative comparisons with other slicing methods interesting and convincing. I have decided to increase my score. I reiterate that I have not checked the mathematical content of this paper in detail.

---

> ### Author Response · Authors · 2020-11-23
> **max-sliced Bures for detecting mode dropping and better reweighting for covariate shift correction**
>
> Thank you for the time and thoroughness. We are excited to report some new results.
>
> Main Concern + Question 1). To highlight how the one-side max-sliced Bures can be used in practice we used it to identify dropped modes in a GAN throughout training. Specifically we use the Stacked MNIST training set and show that the slice that has higher magnitude witness function evaluations in real images versus fake is very precise in retrieving the real images for dropped modes. To be clear, known missing nodes are only needed to quantify the precision of this retrieval. In the case of unknown modes this same approach can be used to identify real-images that are not well matched.
>
> The other slice is useful for finding fake images that are not well-matched. We have included examples from stacked MNIST  in Figure 1 to help explain the approach. Together, examining the witness points from the two slices can provide new insight of coverage of GANs on novel training sets. We believe the approach is complementary to point wise evaluations like Realism scores, or global measures of divergence.
>
> Question 2)  We have now run the results for the different cases from Kolouri et al. (2019): circle, spiral/swiss roll, 8 Gaussians in a ring, and grid of 25 Gaussians.  We also reverted back to the default learning rate as performance was nearly the same. We quantify performance across iterations using the weighted Wasserstein-2 distance, which is solvable as a transportation problem. The max-sliced Bures with appropriate random Fourier bases $\sigma =0.2$ consistently performs the best.
>
> New code release) We have attached the demo and benchmarking scripts for the kernel and linear versions of the max-sliced distances (MATLAB), the MNIST covariate shift detection (MATLAB and requires NIST's EMNIST dataset), the complete code for the Stacked MNIST GAN mode dropping detection experiment (python), and the notebooks and functions for the distribution matching through reweighted using random Fourier basis (extended from Kolouri et al. (2019). For the last, we link an anonymous GitHub repository with a video of training: https://github.com/anon-author-dev/gsw/blob/master/videos/circle_animation.gif).  In the paper, we now include pseudo-code of the one-sided max-sliced Bures divergence.

---

### Decision · Program_Chairs · 2021-01-07
**Final Decision**

**Decision:**

Reject

**Comment:**

The goal in this submission is to find interpretable samples discriminating two probability distributions. In order to tackle this task the authors propose to use a sliced variant of the Bures distance (where the slicing is implemented via a one-rank tensor) and the associated witness function, and illustrate the idea in the discrimination task of fake and real images, and in the detection of covariate shift.

Interpretable discrimination of probability measures with witness functions is a hot topic of machine learning with a large number of applications and available tools (including linear time ones in the sample size, and methods capable of handling independence testing, goodness-of-fit testing, relative tests among others, beyond the considered two sample setting).

1)The motivation of the paper and the efficiency of the proposed method compared to available baselines are not clear; the relevance of the demos is questionable.
2)Unfortunately, the submission also lacks mathematical contributions: for instance
i)Is the proposed divergence a semi-metric or metric, and under what conditions on the domain and the kernel?
ii)Does the proposed estimator converge, under what assumptions, and how quickly (rates)?

The contribution represents a potentially interesting idea, but significantly more work is needed before publication.